# All Convolution, No Attention: Designing Diffusion with Convolutions

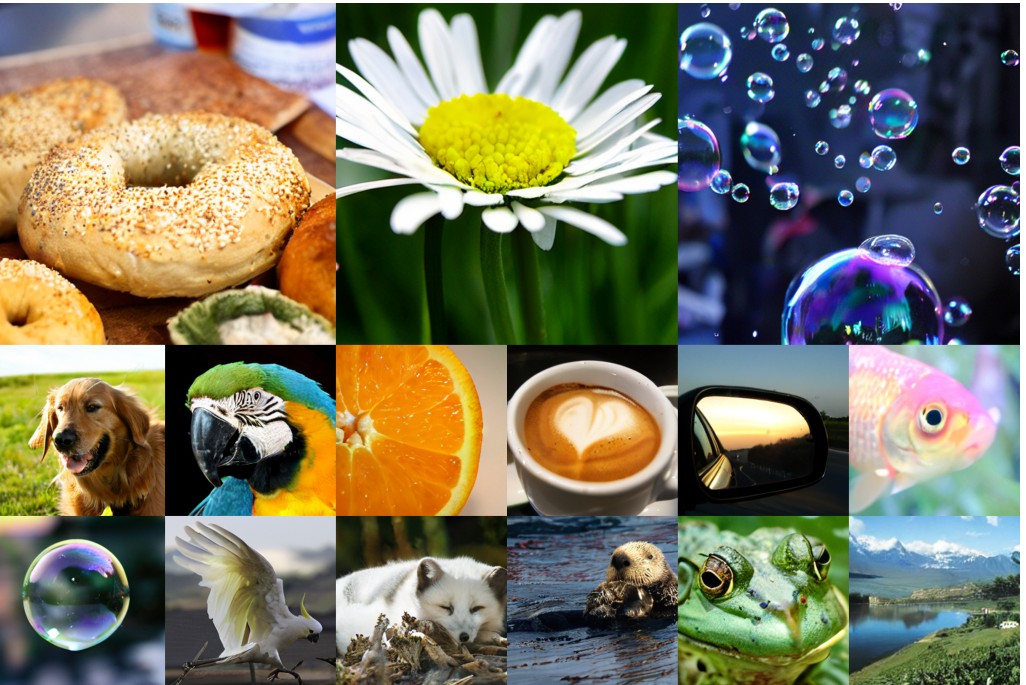

Figure 1: **Diffusion models with fully convolutional backbones achieve high-quality image generation with state-of-the-art efficiency.** We show selected samples from two of our class-conditional FCDM-XL models trained on ImageNet at 512×512 and 256×256 resolution.

## ABSTRACT

Recent diffusion models increasingly favor Transformer backbones, motivated by the remarkable scalability of fully attentional architectures. Yet the locality bias, parameter efficiency, and hardware friendliness—the attributes that established ConvNets as the default vision backbone—have seen limited exploration in modern generative modeling. Here we introduce the *fully convolutional diffusion model (FCDM)*, a ConvNeXt-inspired backbone redesigned for conditional diffusion modeling. Specifically, FCDM employs an easily scalable U-Net hierarchy that integrates global context with fine-grained details and preserves strict convolutional locality, maximizing throughput on modern accelerators. We find that FCDM-XL, using only half the FLOPs of DiT-XL/2, achieves superior FID with 7× and 7.5× speedups at 256×256 and 512×512 resolutions, respectively. Our results demonstrate that modern convolutional designs remain highly competitive when scaled and properly conditioned, challenging the prevailing view that "bigger Transformers" are the sole path to better diffusion models. FCDM revives ConvNets as a compelling, computationally efficient alternative for large-scale generative vision.

## 1 INTRODUCTION

Over the past decade, convolutional neural networks (ConvNets) (LeCun et al., 1989; Krizhevsky et al., 2012; Simonyan & Zisserman, 2015; Szegedy et al., 2015; Ronneberger et al., 2015; He

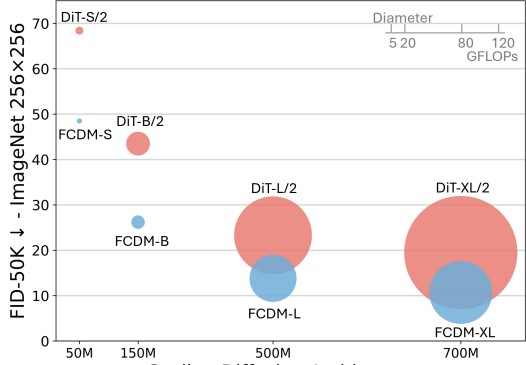

(a) FID and FLOPs comparisons across model scales (400K iterations)

(b) Comparison of FLOPs, throughput, and FID across model scales. The best results are highlighted in **bold**.

| Model | Train Steps | FLOPs (G) | TP ↑ (it/s) | FID ↓ |
|---|---|---|---|---|
| DiT-S/2 | 400K | 6 | 1234 | 68.4 |
| FCDM-S | 400K | **3** | **2687** | **48.5** |
| DiT-B/2 | 400K | 23 | 380.1 | 43.5 |
| FCDM-B | 400K | **12** | **1002** | **26.2** |
| DiT-L/2 | 400K | 81 | 114.6 | 23.3 |
| FCDM-L | 400K | **48** | **381.3** | **13.8** |
| DiT-XL/2 | 400K | 119 | 76.90 | 19.5 |
| FCDM-XL | 400K | **65** | **272.7** | **10.7** |
| DiT-XL/2 | 7M | 119 | 76.90 | 9.6 |
| FCDM-XL | 1M | **65** | **272.7** | **7.9** |

Figure 2: **All Convolution, No Attention.** Is *scalability* exclusive to transformers? Our Fully Convolutional Diffusion Model (FCDM) exhibits clear scalability: it is more efficient and achieves superior performance than Diffusion Transformers (DiTs). Bubble size indicates the FLOPs of each diffusion model. Across all scales (ordered by parameter count), FCDM consistently yields lower FLOPs, higher throughput, and better FID, while converging faster to superior performance.

et al., 2016; Xie et al., 2017; Huang et al., 2017; Howard et al., 2017; Tan & Le, 2019) have driven most major advances in computer vision. Their success stems in part from the implicit "sliding window" mechanism, which embeds a strong locality inductive bias and enables learning effective visual representations with far fewer parameters than fully connected layers. With the incorporation of patch embeddings in the Vision Transformer (ViT) (Dosovitskiy et al., 2021; Liu et al., 2021), Transformers (Vaswani et al., 2017) began to be actively explored in computer vision as well. In particular, the strong scalability of Transformers has allowed them to surpass ConvNets in many areas.

Generative models based on denoising such (Ho et al., 2020; Song et al., 2021a; Dhariwal & Nichol, 2021; Rombach et al., 2022; Karras et al., 2022; Peebles & Xie, 2023; Ma et al., 2024; Esser et al., 2024) have followed similar architectural trends, ranging from hybrid convolution–transformer designs to fully transformer-based backbones. Foundational works (Ho et al., 2020; Song et al., 2021a; Dhariwal & Nichol, 2021; Rombach et al., 2022; Karras et al., 2022) employed a convolutional U-Net architecture augmented with self-attention. DiT (Peebles & Xie, 2023) introduced a fully transformer-based diffusion backbone, replacing convolutions with end-to-end transformer blocks. This shift has driven the success of recent state-of-the-art text-to-image diffusion models (Esser et al., 2024; Labs, 2024), offering improved scalability and generation quality. These developments reflect a prevailing belief that scaling transformer-based networks yields better generative performance. Interestingly, while hybrid convolution-transformer and fully transformer backbones have been extensively studied, fully convolutional backbones for diffusion modeling remain relatively underexplored.

In this work, we revisit the role of convolutions in diffusion modeling. Inspired by ConvNeXt (Liu et al., 2022; Woo et al., 2023), which has demonstrated strong competitiveness with Vision Transformers (Dosovitskiy et al., 2021; Liu et al., 2021) in terms of accuracy and scalability on ImageNet classification (Russakovsky et al., 2015), we design a fully convolutional network tailored for generative diffusion modeling. Specifically, we redesign the ConvNeXt architecture to incorporate conditional injection and organize it in an easy scalable U-shaped design. One of the key contributions of DiT (Peebles & Xie, 2023) is its ease of scaling through a small set of intuitive hyperparameters (e.g., number of blocks $L$, hidden channel $C$, number of heads, and patch size $p$), which has made it highly practical and widely adopted in follow-up research. Our architecture further simplifies this design space, enabling straightforward scaling with only two hyperparameters (number of blocks $L$ and hidden channel $C$). Although built entirely from convolutional modules, the proposed backbone retains the efficiency and scalability of modern ConvNets, benefiting from their fully convolutional nature.

To enable a fair comparison of generation performance among fully convolutional architectures, we train and evaluate our network within the foundational DiT training and evaluation framework (Peebles & Xie, 2023). To assess scalability, we benchmark against DiT models matched in parameter count and find that our model achieves approximately 50% fewer FLOPs. We also observe faster convergence and superior FID performance compared to fully transformer-based architectures. As shown in Figure 2, our **F**ully **C**onvolutional **D**iffusion **M**odel (FCDM) not only is more efficient but also achieves superior performance compared to DiT (Peebles & Xie, 2023) across model scales (ordered by parameter count). These findings re-emphasize the importance of convolutional operations while research increasingly favors Transformer dominance. They also offer a complementary perspective for efficiency-focused work: rather than solely reducing Transformer computational complexity, modern fully convolutional architectures provide an alternative path to scalable, highly efficient generative modeling. We hope these observations and discussions challenge entrenched assumptions and encourage a reevaluation of the role of convolutions in modern computer vision.

## 2 RELATED WORK

This section reviews the architectural evolution of diffusion models, highlighting the transition from convolution–attention hybrid designs to fully transformer-based backbones. From these trends, it is evident that fully convolutional diffusion architectures remain largely underexplored compared to their hybrid and transformer counterparts.

### 2.1 HYBRID ARCHITECTURES

Early diffusion models predominantly adopted hybrid U-Nets (Ronneberger et al., 2015), combining convolutional layers for local features with self-attention (Vaswani et al., 2017) for long-range dependencies. DDPM (Ho et al., 2020), ScoreSDE (Song et al., 2021b), and DDIM (Song et al., 2021a) all used convolutional U-Nets (Ronneberger et al., 2015) augmented with attention at select resolutions. ADM (Dhariwal & Nichol, 2021) further showed that diffusion models could surpass GANs (Goodfellow et al., 2014; Brock et al., 2019; Karras et al., 2019) in high-fidelity image generation, solidifying the architecture's viability. LDM (Rombach et al., 2022) improved scalability by operating in a compressed latent space (Kingma & Welling, 2014), enabling large-scale text-to-image models such as Stable Diffusion and Imagen (Saharia et al., 2022). Recent works including EDM (Karras et al., 2022), EDM2 (Karras et al., 2024), SDXL (Podell et al., 2024), and SnapGen (Chen et al., 2025) refined training strategies and resolution handling while retaining the convolution–attention hybrid backbone.

### 2.2 FULLY TRANSFORMER-BASED DIFFUSION MODELS

Transformers (Vaswani et al., 2017; Dosovitskiy et al., 2021) have emerged as strong alternatives, replacing all convolutions with patch-based attention blocks. DiT (Peebles & Xie, 2023) demonstrated scalability with a ViT-inspired backbone, followed by U-ViT (Bao et al., 2023) and U-DiT (Tian et al., 2024), which introduced U-shaped variants. SiT (Ma et al., 2024) extended DiT to flow matching (Lipman et al., 2023; Liu et al., 2023), surpassing DiT across model scales. PixArt (Chen et al., 2024b;a), MM-DiT (Esser et al., 2024), and FLUX (Labs, 2024) scaled the architecture to production-grade text-to-image models by incorporating improved conditioning pipelines and refined transformer architectures. Recent approaches such as EQVAE (Kouzelis et al., 2025), VAVAE (Yao et al., 2025), and REPA (Yu et al., 2025) further accelerated convergence and improved quality.

### 2.3 FULLY CONVOLUTIONAL DIFFUSION MODELS

Fully convolutional backbones for diffusion models have only recently reemerged. DiC (Tian et al., 2025) re-examined fully convolutional U-Nets using $3\times3$ convolutions with sparse skip connections, while DiCo (Ai et al., 2025) adapted $3\times3$ separable convolutions and proposed compact channel attention to mitigate channel redundancy. Both methods achieve competitive FID with superior throughput, demonstrating their computational efficiency compared to Diffusion Transformers. Our approach diverges from these by adapting ConvNeXt (Liu et al., 2022; Woo et al., 2023), which has demonstrated superior performance to Vision Transformers (ViTs) using only convolutions, and by

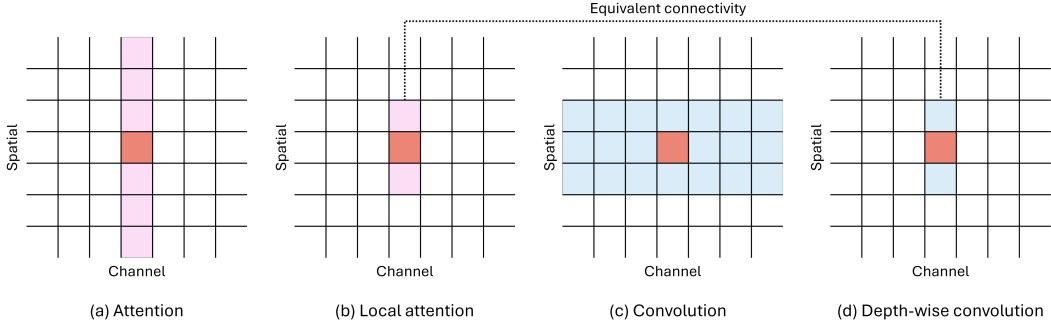

Figure 3: **Illustration of connectivity.** Depthwise convolution is structurally analogous to the weighted-sum operator in self-attention. When attention is restricted to a local window (e.g., local attention), its connectivity becomes identical to that of depthwise convolution.

introducing conditional injection with an easily scalable U-shaped architecture tailored for generative modeling. We demonstrate that our ConvNeXt-inspired backbone not only achieves superior generative performance but also delivers higher computational efficiency than prior fully convolutional diffusion models and Diffusion Transformers, thereby marking a rediscovery of ConvNeXt in the context of generative modeling.

# 3 ALL CONVOLUTION, NO ATTENTION

We propose a **F**ully **C**onvolutional **D**iffusion **M**odel (FCDM), inspired by ConvNeXt (Liu et al., 2022; Woo et al., 2023) and adapted for conditional diffusion generation. Similar to how DiT (Peebles & Xie, 2023) preserves design practices from Vision Transformers (ViTs) (Dosovitskiy et al., 2021), FCDM retains the core principles of ConvNeXt. While ConvNeXt was originally developed for image classification, diffusion modeling imposes distinct requirements. We therefore reassemble ConvNeXt with conditional injection, carefully preserving its core design, and make it a suitable backbone for generative diffusion modeling.

## 3.1 ANALOGY BETWEEN CONVNEXT AND VISION TRANSFORMER

ConvNeXt and Vision Transformers (ViTs) represent two distinct yet structurally analogous in visual representation learning. Transformers rely on self-attention and MLP blocks, while ConvNeXt modernizes convolutional networks with design choices inspired by ViTs (Dosovitskiy et al., 2021; Liu et al., 2021). In the following, we highlight their structural correspondence and explain why ConvNeXt can serve as an effective convolutional alternative to ViTs in visual representation learning.

**Depthwise convolution has the same connectivity as local attention.** Depthwise convolution (Chollet, 2017; Howard et al., 2017) is structurally analogous to the weighted-sum operator in self-attention (Liu et al., 2022). As illustrated in Figure 3, when attention is restricted to a local window, its connectivity becomes identical to that of depthwise convolution: each spatial position connects only to its local neighborhood without cross-channel mixing (Han et al., 2022).

Local Vision Transformers (Local ViTs), such as Swin Transformer (Liu et al., 2021) and HaloNet (Vaswani et al., 2021), employ local attention to improve both accuracy and efficiency in image classification compared to the standard ViT (Dosovitskiy et al., 2021). With the same local connectivity, as illustrated in Figure 3, depthwise convolution–based networks such as ConvNeXt (Liu et al., 2022) and DWNet (Han et al., 2022) have demonstrated that their architectures can match or even surpass Local ViTs on the ImageNet classification benchmark. In particular, ConvNeXt shows that standard depthwise convolution with shared kernels is sufficient to outperform Local ViTs once the connectivity is equivalent (Liu et al., 2022). Building on this insight, we adopt depthwise convolution with shared kernels as a simple and effective design choice, rather than relying on more complex dynamic weighting (Han et al., 2022).

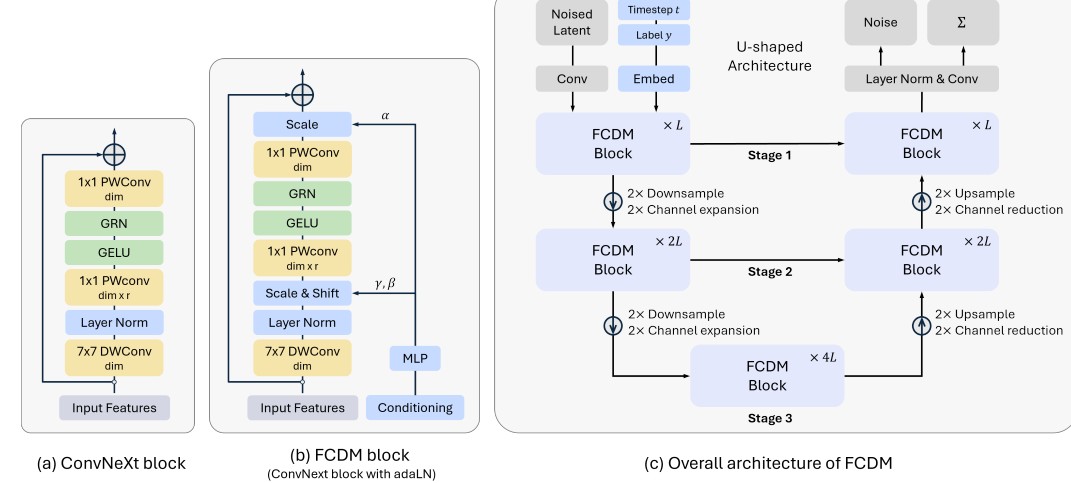

(a) ConvNeXt block

(b) FCDM block
(ConvNext block with adaLN)

(c) Overall architecture of FCDM

Figure 4: **The Fully Convolutional Diffusion Model (FCDM) architecture.** (a) Details of the ConvNeXt block. (b) Our FCDM block, which incorporates conditioning via adaptive layer normalization. (c) We train conditional latent FCDMs. The input latent is processed by multiple FCDM blocks arranged in an easily scalable U-shaped architecture.

| Model | Blocks $L$ | Hidden channel $C$ | Params (M) | FLOPs (G) | $\frac{\text{FLOPs (FCDM)}}{\text{FLOPs (DiT)}}$ |
|---|---|---|---|---|---|
| FCDM-S | 2 | 128 | 32.7 | 3.1 | 50.8% |
| FCDM-B | 2 | 256 | 127.7 | 12.2 | 53.0% |
| FCDM-L | 2 | 512 | 504.5 | 48.3 | 59.9% |
| FCDM-XL | 3 | 512 | 698.8 | 64.6 | 54.5% |

Table 1: We follow parameter counts of Diffusion Transformers (DiTs) for the Small (S), Base (B), Large (L), and XLarge (XL) scales. FLOPs are measured on ImageNet 256×256.

**Pointwise convolutions are equivalent to Transformer MLPs.** Following depthwise convolution, ConvNeXt applies 1×1 pointwise convolutions in an *inverted bottleneck* structure, which mix information across channels at each spatial location. This operation is mathematically equivalent to the Transformer MLP: the channel dimension is first expanded by a ratio $r$ and then projected back using two linear layers, where a nonlinear activation is applied between them. From this perspective, 1×1 pointwise convolutions fulfill the same structural role as MLP blocks in ViTs, but through purely convolutional operations.

With these correspondences, ConvNeXt exhibits a clear structural alignment with ViTs (Dosovitskiy et al., 2021; Liu et al., 2021) and has demonstrated superior performance on ImageNet classification (Russakovsky et al., 2015). This shows that convolutional architectures can retain the benefits of Transformer connectivity while remaining simpler and more efficient.

## 3.2 DESIGNING DIFFUSION WITH CONVOLUTIONS

Building on the structural correspondences between ConvNeXt and ViTs, we redesign ConvNeXt into a generative backbone for diffusion models, introducing Fully Convolutional Diffusion Model (FCDM). In this section, we highlight the advantages of fully convolutional architectures compared to Transformer-based designs, and describe the key components that define the design space of the FCDM class.

**Conditional injection.** Original ConvNeXt blocks lack conditioning mechanisms as shown in Figure 4 (a). To enable class and time conditioning, we replace LayerNorm with Adaptive LayerNorm (AdaLN), as shown in Figure 4 (b). A lightweight MLP maps the conditioning vector (derived from class and time embeddings) to $(\gamma, \beta, \alpha)$ parameters that modulate normalized features. Following DiT (Peebles & Xie, 2023), we zero-initialize the final modulation scale $\alpha$ to stabilize optimization and allow deeper training.

**Easily scalable U-shaped architecture.** Most convolutional networks adopt a U-shaped design with skip connections, which facilitates the integration of global and local features. This structure makes it easier to capture the overall context while preserving the high-resolution details from the early encoder layers. Following this principle, we organize ConvNeXt blocks within a U-Net hierarchy, with skip connections bridging the encoder and decoder stages.

To simplify scalability, we avoid the complex, resolution-specific design choices often used in U-shaped networks. Instead, our architecture is parameterized by only two hyperparameters: the block count $L$ and the number of hidden channels $C$. At each $2\times$ downsampling stage, both $C$ and $L$ are doubled. This *generalized U-shaped* design (Figure 4 (c)) allows straightforward scaling while retaining the inductive biases of convolutions. By controlling $C$ and $L$, the proposed architecture can be scaled up or down in a straightforward manner.

## 4 EXPERIMENTAL SETUP

**Model size.** We denote our models by their configurations, parameterized by hidden channels $C$ and number of blocks $L$, which are both double at each $2\times$ downsampling stage. To enable fair comparisons, we align the parameter counts of our FCDM scales with those of DiT (Peebles & Xie, 2023) (e.g., DiT-B: 130M vs. FCDM-B: 127.7M). We evaluate four model scales, as listed in Table 1: FCDM-S, FCDM-B, FCDM-L, and FCDM-XL. These cover a broad range of number of parameters, from 32.7M to 698.8M, allowing us to systematically study scaling behavior and compare with DiT across different scales.

**Training.** We train class-conditional latent FCDMs at $256\times256$ and $512\times512$ resolutions on the ImageNet dataset (Russakovsky et al., 2015), a standard yet highly competitive benchmark for generative modeling. Training follows common practices of DiT (Peebles & Xie, 2023): we use AdamW (Kingma & Ba, 2015; Loshchilov & Hutter, 2019) with a fixed learning rate of $1 \times 10^{-4}$, no weight decay, and batch size 256. The only augmentation applied is horizontal flipping. We use an exponential moving average (EMA) of model weights with a decay factor of 0.9999, and report all results using the EMA weights. We retain diffusion hyperparameters from ADM (Dhariwal & Nichol, 2021): $t_{\max} = 1000$ steps with a linear variance schedule ($1 \times 10^{-4}$ to $2 \times 10^{-4}$), ADM's co-variance parameterization $\Sigma_\theta$, and their timestep/label embedding method. See Appendix A for an overview of denoising diffusion probabilistic models and Appendix B for additional training details and hyperparameters.

**Datasets and Metrics.** We conduct experiments on ImageNet-1K at $256\times256$ and $512\times512$ resolutions for class-conditional image generation. Our primary metric is Fréchet Inception Distance (FID) (Heusel et al., 2017), following the standard evaluation protocol. We sample 50K images with 250 DDPM sampling steps, and compute the metrics using OpenAI's official TensorFlow evaluation toolkit (Dhariwal & Nichol, 2021). As secondary metrics, we also report Inception Score (IS) (Salimans et al., 2016) and Precision/Recall (Kynkäänniemi et al., 2019). See Appendix C for more details.

**Compute.** All models are implemented in PyTorch (Paszke et al., 2019) and trained on a cluster of RTX 4090 GPUs. The largest model, FCDM-XL, trains at roughly 0.9 iterations per second (with gradient checkpointing) on $256\times256$ training, using only 4 NVIDIA RTX 4090 24GB GPUs with a global batch size of 256.

## 5 EXPERIMENTS

### 5.1 SCALING MODEL SIZE

We train four FCDM models (S, B, L, XL), all using the same training configuration. Figure 2 (a) summarizes the FLOPs and FID at 400K training iterations. In all cases, scaling up model size improves performance. Figure 5 further shows that FCDM consistently outperforms DiT (Peebles & Xie, 2023) across all scales. Increasing model scale, both in width and depth, consistently leads to significant FID improvements.

| Model | Training iterations | FLOPs (G) ↓ | Throughput (it/s) ↑ | FID ↓ | IS ↑ | Precision ↑ | Recall ↑ |
|---|---|---|---|---|---|---|---|
| DiT-S/2 *(ICCV 2023)* | 400K | 6.1 | 1234.0 | 68.40 | - | - | - |
| DiC-S *(CVPR 2025)* | 400K | 5.9 | **3148.8** | 58.68 | 25.82 | - | - |
| DiG-S/2 *(CVPR 2025)* | 400K | 4.3 | 961.2 | 62.06 | 22.81 | 0.39 | 0.56 |
| DiCo-S *(NeurIPS 2025)* | 400K | 4.3 | 1695.7 | 49.97 | 31.38 | **0.48** | **0.58** |
| **FCDM-S** | 400K | **3.1** | 2687.2 | **48.52** | **31.64** | **0.48** | **0.58** |
| DiT-B/2 | 400K | 23.0 | 380.1 | 43.47 | - | - | - |
| DiC-B | 400K | 23.5 | **1024.2** | 32.33 | 48.72 | - | - |
| DiG-B/2 | 400K | 17.1 | 345.9 | 39.50 | 37.21 | 0.51 | **0.63** |
| DiCo-B | 400K | 16.9 | 823.0 | 27.20 | 56.52 | **0.60** | 0.61 |
| **FCDM-B** | 400K | **12.2** | 1001.6 | **26.21** | **58.04** | 0.59 | 0.61 |
| DiT-L/2 | 400K | 80.7 | 114.6 | 23.33 | - | - | - |
| DiG-L/2 | 400K | 61.7 | 109.0 | 22.90 | 59.87 | 0.60 | **0.64** |
| DiCo-L | 400K | 60.2 | 288.3 | **13.66** | 91.37 | **0.69** | 0.61 |
| **FCDM-L** | 400K | **48.3** | **381.3** | 13.83 | 93.31 | 0.66 | 0.62 |
| DiT-XL/2 | 400K | 118.6 | 76.9 | 19.47 | - | - | - |
| DiC-XL | 400K | 116.1 | 263.1 | 13.11 | 100.2 | - | - |
| DiG-XL/2 | 400K | 89.4 | 71.7 | 18.53 | 68.53 | 0.63 | **0.64** |
| DiCo-XL | 400K | 87.3 | 208.5 | 11.67 | 100.4 | **0.71** | 0.61 |
| DiC-H | 400K | 204.4 | 144.5 | 11.36 | 106.5 | - | - |
| **FCDM-XL** | 400K | **64.6** | 272.7 | **10.72** | **108.0** | 0.69 | 0.63 |
| DiT-XL/2 | 7M | 118.6 | 76.9 | 9.62 | - | - | - |
| DiC-H | 800K | 204.4 | 144.5 | 8.96 | 124.33 | - | - |
| DiG-XL/2 | 1.2M | 89.4 | 71.7 | 8.60 | 130.03 | 0.68 | **0.68** |
| **FCDM-XL** | 1M | **64.6** | 272.7 | **7.91** | **135.55** | **0.71** | 0.64 |

Table 2: **Scalability comparisons on ImageNet 256×256.** For each model scale, we report FID, IS, Precision, and Recall (50K samples without guidance), and efficiency metrics (training iterations, FLOPs, throughput). FCDM-XL achieves superior convergence while using 50% fewer FLOPs than DiT-XL/2. Reported values follow the respective papers; the best results are highlighted in **bold**.

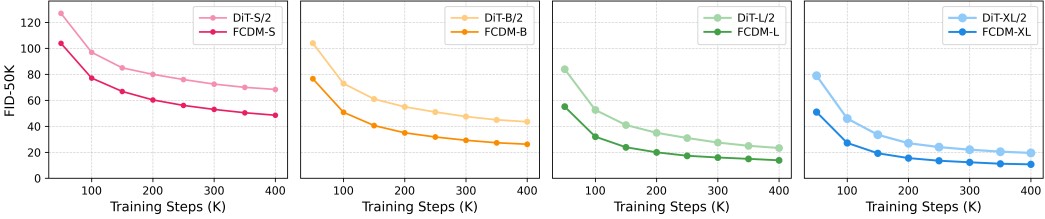

Figure 5: **FCDM improves FID across all model scales.** FID-50K over training iterations for both DiT and FCDM. Across all model scales, FCDM converges much faster.

Table 2 provides a broader comparison with DiT and recent state-of-the-art class-conditional models that follow a similar experimental setup. Although our scales are aligned with DiT in terms of parameter counts, FCDM requires about 50% fewer FLOPs than DiT and 30% fewer FLOPs than DiCo (Ai et al., 2025). Notably, FCDM-XL (64.6G FLOPs) is computationally closer to Large (L) scale models in prior works, yet outperforms even XL-scale models in terms of FID. In particular, FCDM-XL achieves superior FID with a 7× speedup compared to DiT-XL/2. Furthermore, its efficiency also yields favorable throughput: while DiC (Tian et al., 2025) has the best throughput at S and B scales due to its use of standard convolution, which is simpler and better supported by hardware, FCDM outperforms it at L and XL scales, achieving the fastest throughput. For completeness, detailed descriptions of each baseline method are provided in Appendix D, and a detailed scaling analysis is provided in Appendix E.

## 5.2 BENCHMARKING PERFORMANCE AND EFFICIENCY

**256×256 ImageNet.** Building on the scaling analysis, we train FCDM-XL for 2M iterations (400 epochs) and evaluate it with classifier-free guidance (Ho & Salimans, 2021). Figure 1 presents generated samples and Table 3 further compares against prior class-conditional image generation models. Notably, FCDM-XL improves upon baselines despite requiring fewer training epochs. In particular, it achieves an FID of 2.03 and an IS of 285.7, while reaching state-of-the-art efficiency in FLOPs and throughput, demonstrating a strong trade-off between performance and efficiency. As shown in Figure 6, our model achieves state-of-the-art throughput while drastically reducing training cost.

| Model | Training epochs | FLOPs (G) ↓ | Throughput (it/s) ↑ | FID ↓ | IS ↑ | Precision ↑ | Recall ↑ |
|---|---|---|---|---|---|---|---|
| *GAN* | | | | | | | |
| BigGAN-deep | - | - | - | 6.95 | 171.4 | **0.87** | 0.28 |
| StyleGAN-XL | - | - | - | 2.30 | 265.12 | 0.78 | 0.53 |
| *Pixel diffusion* | | | | | | | |
| ADM-U | 400 | 742.0 | - | 3.94 | 215.8 | 0.83 | 0.53 |
| VDM++ | 560 | - | - | 2.12 | 267.7 | - | - |
| Simple Diffusion | 800 | - | - | 2.77 | 211.8 | - | - |
| CDM | 2160 | - | - | 4.88 | 158.7 | - | - |
| *Latent diffusion* | | | | | | | |
| LDM-4 | 200 | 104.0 | - | 3.60 | 247.7 | **0.87** | 0.48 |
| U-ViT-H/2 | 240 | 133.3 | 73.5 | 2.29 | 263.9 | 0.82 | 0.57 |
| MaskDiT | 1600 | - | - | 2.28 | 276.6 | 0.80 | **0.61** |
| SD-DiT | 480 | - | - | 3.23 | - | - | - |
| DiT-XL/2 | 1400 | 118.6 | 76.9 | 2.27 | 278.2 | 0.83 | 0.57 |
| SiT-XL/2 | 1400 | 118.6 | 76.9 | 2.06 | 277.5 | 0.83 | 0.59 |
| DiG-XL/2 | 240 | 89.4 | 71.7 | 2.07 | 279.0 | 0.82 | 0.60 |
| DiCo-XL | 750 | 87.3 | 174.2 | 2.05 | 282.2 | 0.83 | 0.59 |
| DiC-H | 400 | 204.4 | 144.5 | 2.25 | - | - | - |
| **FCDM-XL** | 400 | **64.6** | **272.7** | **2.03** | **285.7** | 0.81 | 0.59 |

Table 3: **Benchmarking class-conditional image generation on ImageNet 256×256.** We compare representative models in terms of FID, IS, Precision, Recall (with guidance), and efficiency metrics (training epochs, FLOPs, throughput). FCDM-XL achieves both superior efficiency and performance. Reported values follow the respective papers; the best results are highlighted in **bold**.

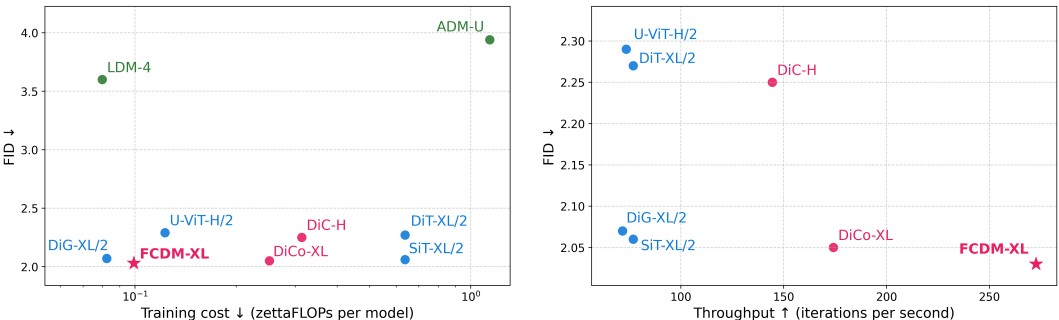

Figure 6: **Benchmarking class-conditional image generation performance and efficiency on ImageNet 256×256.** *Left*: FID versus training cost. *Right*: FID versus throughput. One zettaFLOP corresponds to $10^{21}$ FLOPs ($10^{12}$ GFLOPs). A training iteration is assumed to cost about 3× one evaluation (forward + backward to inputs + backward to weights). Red denotes fully convolutional, Green denotes hybrid, and Blue denotes fully transformer-based models.

These results highlight, for the first time, the effectiveness of ConvNeXt architectures in generative diffusion modeling, which had previously been shown only in classification, and demonstrate their broader adaptability and potential. It is important to note that, although FCDM-XL demonstrates superior performance over comparable baselines within DiT-style experimental settings, it does not yet surpass the latest state-of-the-art results achieved by models such as EDM-2 (Karras et al., 2024) or Simpler Diffusion (Hoogeboom et al., 2025). Nevertheless, as an architecture with a favorable performance–efficiency trade-off, our approach holds the potential to deliver improved results with further scaling and more advanced training frameworks.

**512×512 ImageNet.** We also train FCDM-XL at 512×512 resolution for 1M iterations, using the same hyperparameters as the 256×256 model. Table 4 reports FLOPs and FID at both 400K and 1M iterations. At this resolution, FCDM-XL once again achieves the best FLOPs and throughput. With this efficiency advantage, FCDM-XL obtains a superior FID at 400K iterations, which further persists at 1M. Remarkably, FCDM-XL outperforms DiT and DiCo even with far fewer training iterations, achieving a 7.5× speedup over DiT-XL/2 and a 3× speedup over DiCo. These results demonstrate that FCDM not only maintains strong efficiency but also exceeds models trained with substantially more iterations, confirming the superiority of the architecture. Interestingly, when resolution doubles, throughput of DiT drops by 4×, while FCDM drops by only 2×. This contrast highlights the computational differences between fully transformer-based and fully convolutional-

| Model | Training iterations | FLOPs (G) ↓ | Throughput (it/s) ↑ | FID ↓ | IS ↑ | Precision ↑ | Recall ↑ |
|---|---|---|---|---|---|---|---|
| DiT-XL/2 | 400K | 524.7 | 18.6 | 20.94 | 66.3 | 0.74 | 0.58 |
| DiG-XL/2 | 400K | - | - | 17.36 | 69.4 | 0.75 | **0.63** |
| DiC-XL | 400K | 464.3 | 124.2 | 15.32 | 93.6 | - | - |
| DiC-H | 400K | 817.2 | 68.6 | 12.89 | 101.8 | - | - |
| **FCDM-XL** | 400K | **257.7** | **129.6** | **10.23** | **108.7** | **0.79** | 0.60 |
| DiT-XL/2 | 3M | 524.7 | 18.6 | 12.03 | 105.3 | 0.75 | **0.64** |
| DiCo-XL | 3M | 349.8 | 82.0 | 7.48 | **146.4** | 0.78 | 0.63 |
| **FCDM-XL** | 1M | **257.7** | **129.6** | **7.46** | 133.6 | **0.79** | 0.61 |

Table 4: **Benchmarking class-conditional image generation on ImageNet 512×512.** We report FID, IS, Precision, Recall (without guidance), and efficiency metrics for representative models. Even at this resolution, FCDM surpasses models trained for 3M iterations with only 1M iterations and achieves best efficiency in FLOPs and throughput. The best results are highlighted in **bold**.

| Model | Training iterations | FLOPs (G) ↓ | FID ↓ | IS ↑ | Precision ↑ | Recall ↑ |
|---|---|---|---|---|---|---|
| **FCDM-L** (Default: 7×7 DWConv) | 200K | 48.3 | **19.97** | 69.19 | **0.6312** | **0.6128** |
| 7×7 → 5×5 DWConv | 200K | 48.2 | 20.48 | 66.69 | 0.6310 | 0.6017 |
| 7×7 → 3×3 DWConv | 200K | 48.1 | 21.28 | 64.11 | 0.6269 | 0.5993 |
| **FCDM-L** (Default: FCDM block) | 200K | 48.3 | **19.97** | 69.19 | **0.6312** | **0.6128** |
| FCDM block → ResNet block* | 200K | 48.4 | 31.14 | 49.10 | 0.5866 | 0.5926 |
| **FCDM-L** (Default: U-shaped) | 200K | 48.3 | **19.97** | 69.19 | **0.6312** | **0.6128** |
| U-shaped → Isotropic | 200K | 46.1 | 41.15 | 33.97 | 0.5504 | 0.5889 |
| **FCDM-L** (Default: w/ GRN) | 200K | 48.3 | **19.97** | 69.19 | **0.6312** | **0.6128** |
| w/o GRN | 200K | 48.2 | 21.24 | 62.35 | 0.6302 | 0.5923 |

Table 5: **Ablation study on design choices of FCDM.** We analyze the effects of kernel size, FCDM block, U-shaped architecture, and GRN. * indicates that the hidden channel is adjusted (512 → 336) to match FLOPs for fair comparison.

based designs, and demonstrates the superior suitability of FCDM for scaling to higher resolutions. Moreover, we provide a frequency-based analysis in Appendix F as an observation to better understand the model behavior in comparison with Diffusion Transformers, and additional qualitative examples in Appendix H.

## 5.3 ABLATION STUDY

We conduct ablations on the Large (L) model at 256×256 ImageNet to analyze the effect of key architectural components, as summarized in Table 5. Reducing kernel size consistently degrades performance, demonstrating that large kernels strengthen local operations and effectively approximate global context. Replacing FCDM blocks with ResNet blocks (He et al., 2016) results in a severe degradation, highlighting the advantage of our ConvNeXt-inspired design. Removing the U-shaped hierarchy and adopting an isotropic architecture similarly degrades performance, showing the importance of combining local detail with global context. Finally, eliminating GRN results in a clear performance drop, confirming its role in reducing channel redundancy and showing that it achieves a similar effect to compact channel attention of DiCo but with far fewer parameters. Detailed and additional ablation results are provided in Appendix G.

## 6 CONCLUSION

In this work, we revisited the role of convolutions in diffusion modeling and introduced FCDM, a ConvNeXt-inspired backbone tailored for generative tasks. By combining an easily scalable U-shaped hierarchy with strict convolutional locality, FCDM achieves superior generative performance and higher computational efficiency than prior fully convolutional designs and even Diffusion Transformers. These results demonstrate that modern convolutional architectures, when carefully adapted, remain highly competitive and challenge the prevailing belief that larger Transformers are the sole path to progress in diffusion models. We provide a discussion of future research directions in Appendix I.

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

APPENDIX

We provide the following supplementary materials in the Appendix:

- Section A: Overview of denoising diffusion probabilistic models (DDPMs).
- Section B: Hyperparameter and implementation details.
- Section C: Descriptions of evaluation metrics.
- Section D: Descriptions of baseline models.
- Section E: Additional results and analyses on model scalability.
- Section F: Frequency-based analysis of model behavior.
- Section G: Additional quantitative analyses on architectural variants.
- Section H: Additional qualitative results and visual samples.
- Section I: Discussions on potential directions for future research.

## A  OVERVIEW ON DENOISING DIFFUSION PROBABILISTIC MODELS

Diffusion models (Sohl-Dickstein et al., 2015; Ho et al., 2020) aim to model a target distribution $p(x)$ by learning a gradual denoising process from Gaussian noise $\mathcal{N}(0, I)$ to $p(x)$. Specifically, the model learns a *reverse* process $p_\theta(x_{t-1}|x_t)$ of a predefined *forward* diffusion process $q(x_t|x_{t-1})$, which progressively adds Gaussian noise over $T$ timesteps.

For an initial sample $x_0 \sim p(x)$, the *forward* process is defined as:

$$q(x_t|x_{t-1}) = \mathcal{N}\Big(x_t; \sqrt{1 - \beta_t}\, x_{t-1}, \beta_t I\Big), \tag{1}$$

where $\beta_t \in (0, 1)$ is a variance schedule. A closed-form expression of $q(x_t|x_0)$ can also be derived as:

$$q(x_t|x_0) = \mathcal{N}\Big(x_t; \sqrt{\bar{\alpha}_t}\, x_0, (1 - \bar{\alpha}_t)I\Big), \quad \bar{\alpha}_t = \prod_{s=1}^{t}(1 - \beta_s). \tag{2}$$

Denoising Diffusion Probabilistic Model (DDPM) (Ho et al., 2020) parameterizes the *reverse* transition as

$$p_\theta(x_{t-1}|x_t) = \mathcal{N}\Big(x_{t-1}; \frac{1}{\sqrt{\alpha_t}}\Big(x_t - \frac{1 - \alpha_t}{\sqrt{1 - \bar{\alpha}_t}}\, \epsilon_\theta(x_t, t)\Big), \sigma_t^2 I\Big), \tag{3}$$

where the noise predictor $\epsilon_\theta(x_t, t)$ is trained using a simple denoising autoencoder objective:

$$\mathcal{L}_{\text{simple}} = \mathbb{E}_{x_t, x_0, \epsilon, t}\Big[\|\epsilon - \epsilon_\theta(x_t, t)\|_2^2\Big]. \tag{4}$$

Following DDPM, one can set $\sigma_t^2 = \beta_t$ for simplicity. Meanwhile, improved DDPM (iDDPM) (Nichol & Dhariwal, 2021) shows that performance can be improved by jointly learning the variance $\Sigma_\theta(x_t, t)$, which is parameterized as an interpolation between $\beta_t$ and $\tilde{\beta}_t$ in the log domain:

$$\log \Sigma_\theta(x_t, t) = v \log \beta_t + (1 - v) \log \tilde{\beta}_t, \tag{5}$$

where $\tilde{\beta}_t = \frac{1 - \bar{\alpha}_{t-1}}{1 - \bar{\alpha}_t}\beta_t$, and $v$ denotes the interpolation weight predicted by the model in a dimension-wise manner. In this work, we adopt the iDDPM framework for both training and sampling, following the same design choice as DiT (Peebles & Xie, 2023).

## B  HYPERPARAMETERS AND IMPLEMENTATION DETAILS

We design Fully Convolutional Diffusion Models (FCDMs) at multiple scales, aligned by parameter counts with DiT. Thanks to their easy scalability, we adjust only two hyperparameters, $L$ and $C$, to obtain these variants. Notably, when compared to DiT in terms of FLOPs, our FCDMs require only $50.8\%$ to $59.9\%$ of the FLOPs consumed by DiT, demonstrating the state-of-the-art computational efficiency of our design. Our implementation is based on the original DiT codebase (Peebles & Xie, 2023). Detailed configurations of these hyperparameters, along with additional implementation details, are provided in Table 6. For the latent space, we adopt an off-the-shelf pre-trained variational autoencoder (VAE) (Kingma & Welling, 2014; Rombach et al., 2022; Kouzelis et al., 2025) with a downsampling factor of 8. Accordingly, an input RGB image of shape $256\times256\times3$ is encoded to a latent tensor of $32\times32\times4$. All diffusion training operates in this latent space, and latents are decoded back to pixels by the VAE decoder.

|  | FCDM-S | FCDM-B | FCDM-L | FCDM-XL | FCDM-XL |
| --- | --- | --- | --- | --- | --- |
| Resolution | 256×256 | 256×256 | 256×256 | 256×256 | 512×512 |
| **Architecture** | | | | | |
| Input dim. | 32×32×4 | 32×32×4 | 32×32×4 | 32×32×4 | 64×64×4 |
| Num. blocks ($L$) | 2 | 2 | 2 | 3 | 3 |
| Hidden channels ($C$) | 128 | 256 | 512 | 512 | 512 |
| 1×1 conv. expansion ratio ($r$) | 3 | 3 | 3 | 3 | 3 |
| **Training** | | | | | |
| Training iteration | 400K | 400K | 400K | 2M | 1M |
| Global batch size | 256 | 256 | 256 | 256 | 256 |
| Optimizer | AdamW | AdamW | AdamW | AdamW | AdamW |
| Learning rate | $1 \times 10^{-4}$ | $1 \times 10^{-4}$ | $1 \times 10^{-4}$ | $1 \times 10^{-4}$ | $1 \times 10^{-4}$ |
| Learning rate schedule | constant | constant | constant | constant | constant |
| $(\beta_1, \beta_2)$ | (0.9, 0.999) | (0.9, 0.999) | (0.9, 0.999) | (0.9, 0.999) | (0.9, 0.999) |
| Weight decay | 0 | 0 | 0 | 0 | 0 |
| Numerical precision | fp32 | fp32 | fp32 | fp32 | fp32 |
| Data augmentation | random flip | random flip | random flip | random flip | random flip |
| **Sampling** | | | | | |
| Sampler | iDDPM | iDDPM | iDDPM | iDDPM | iDDPM |
| Sampling steps | 250 | 250 | 250 | 250 | 250 |

Table 6: **Hyperparameter setup of FCDM model scales.** For all scales of FCDM, we adopt the same experimental settings as DiT.

**Computing resources.**  Thanks to the superior efficiency of our fully convolutional architecture, we are able to train $256 \times 256$ ImageNet models even with *consumer*-level GPUs such as NVIDIA RTX 4090 24GB. Specifically, for $256 \times 256$ resolution, we use 4 RTX 4090 24GB GPUs with a training speed of about 0.9 steps/s (with gradient checkpointing) for FCDM-XL at a batch size of 256. For $512 \times 512$, we use 4 NVIDIA H100 80GB GPUs, achieving a training speed of 0.7 steps/s (with gradient checkpointing) with the same batch size.

## C  EVALUATION METRICS

For evaluation, we follow the setup of ADM (Dhariwal & Nichol, 2021) and use the same reference batches provided in their official implementation.[1] Specifically, we generate 50K samples and compute the metrics using OpenAI's official TensorFlow evaluation toolkit. All evaluations are conducted on NVIDIA RTX 4090 or NVIDIA H100 GPUs, except for certain reported numbers that are taken from prior work.

The following gives a concise description of the evaluation metrics used in our experiments.

---

[1] https://github.com/openai/guided-diffusion/tree/main/evaluations

- **FID** (Heusel et al., 2017) measures the distance between the feature distributions of real and generated images. It is computed using the Inception-v3 network (Szegedy et al., 2016), under the assumption that both feature distributions follow multivariate Gaussian distributions.

- **sFID** (Nash et al., 2021) computes FID using spatial feature maps from intermediate layers of Inception-v3, thereby better capturing the spatial structure of generated images.

- **IS** (Salimans et al., 2016) evaluates only generated images using the Inception-v3 network. It assigns higher scores when the images are classifiable with high confidence (sharp and meaningful) and when the set of generated images is diverse across different categories.

- **Precision and Recall** (Kynkäänniemi et al., 2019) measure realism and diversity in feature space. Precision reflects the fraction of generated images that look realistic, while recall reflects how much of the real data distribution is covered by the generated samples.

We additionally report computational efficiency. FLOPs are computed using torchprofile[2], and throughput is evaluated under the sampling configurations of DiT (Peebles & Xie, 2023) with a batch size of 64. FlashAttention-2 (Dao, 2024) and Flash Linear Attention (Yang et al., 2024) are activated in DiT and DiG, respectively.

## D  BASELINE MODELS

The following summarizes the key ideas of the diffusion baselines used for the evaluation.

- **ADM** (Dhariwal & Nichol, 2021) improves hybrid U-Net architecture for diffusion models and introduces classifier guidance, which enables a trade-off between sample quality and diversity.

- **VDM++** (Kingma & Gao, 2023) enhances training efficiency by proposing a simple adaptive noise schedule for diffusion models.

- **Simple diffusion** (Hoogeboom et al., 2023) proposes a diffusion model for high-resolution image generation by carefully redesigning the noise schedule and model architecture.

- **CDM** (Ho et al., 2022) adopts a cascaded framework in which a base model first generates a low-resolution image, and subsequent super-resolution diffusion models progressively refine it to higher fidelity.

- **LDM** (Rombach et al., 2022) proposes latent diffusion models that operate in a compressed latent space, greatly improving training efficiency while retaining high generation quality.

- **U-ViT** (Bao et al., 2023) adapts Vision Transformers for latent diffusion by introducing long skip connections similar to those in U-Net.

- **MaskDiT** (Zheng et al., 2024) proposes an asymmetric encoder–decoder architecture for diffusion transformers, trained with an auxiliary mask reconstruction task to improve efficiency.

- **SD-DiT** (Zhu et al., 2024) reframes the mask modeling of MaskDiT as a self-supervised discrimination objective.

- **DiT** (Peebles & Xie, 2023) replaces the hybrid U-Net architecture with a fully transformer-based backbone, introduces AdaIN-zero conditioning to stabilize training, and shows that diffusion transformers scale effectively.

- **SiT** (Ma et al., 2024) reformulates DiT training by transitioning from discrete diffusion to continuous flow matching.

- **DiG** (Zhu et al., 2025) integrates Gated Linear Attention (Yang et al., 2024), enabling sub-quadratic complexity efficiency of diffusion transformers.

- **DiC** (Tian et al., 2025) re-examined purely convolutional denoisers by scaling standard 3×3 convolutional blocks in a U-Net design, introducing sparse skip connections.

- **DiCo** (Ai et al., 2025) proposes 3×3 seperable convolutional block in a U-Net design, introducing compact chennal attention to activate more informative channels.

---

[2]https://pypi.org/project/torchprofile/

# E ADDITIONAL SCALING RESULTS

As shown in Figure 5, we clearly demonstrate the scalability of FCDM in terms of FID. We also observe consistent scalability across other metrics, including sFID, Inception Score, Precision, and Recall, as reported in Table 7.

In addition, we trained FCDMs using the original SiT implementation (Ma et al., 2024) to examine whether our proposed design also exhibits scalability under this framework. Following the original implementation details, we trained the model with the flow-matching objective (Lipman et al., 2023; Ma et al., 2024). We used AdamW with a constant learning rate of $1\times10^{-4}$, $(\beta_1, \beta_2) = (0.9, 0.999)$, and no weight decay. For sampling, we employed the Euler–Maruyama SDE sampler with 250 steps, setting the final step size to 0.04.

As shown in Table 8, we again observe clear scalability when training the proposed network with the flow-matching objective. Interestingly, under our framework, the flow-matching objective yields better performance at the Small (S) scale, while showing slightly worse results at the Base (B) through XLarge (XL) scales. Nevertheless, these results confirm that the proposed architecture possesses generalized scalability beyond a specific training objective.

| Model | FLOPs (G) | Training Steps | FID ↓ | sFID ↓ | IS ↑ | Precision ↑ | Recall ↑ |
|-------|-----------|----------------|-------|--------|------|-------------|----------|
| FCDM-S | 3.10 | 50K | 103.93 | 15.03 | 12.01 | 0.3013 | 0.3513 |
| | | 100K | 77.17 | 12.15 | 17.11 | 0.3809 | 0.4473 |
| | | 150K | 66.85 | 11.02 | 20.68 | 0.4136 | 0.5106 |
| | | 200K | 60.33 | 10.70 | 23.71 | 0.4396 | 0.5537 |
| | | 250K | 56.08 | 10.54 | 26.30 | 0.4568 | 0.5609 |
| | | 300K | 53.01 | 10.59 | 28.10 | 0.4687 | 0.5806 |
| | | 350K | 50.44 | 10.29 | 30.08 | 0.4757 | 0.5729 |
| | | 400K | **48.53** | **10.12** | **31.64** | **0.4836** | **0.5840** |
| FCDM-B | 12.20 | 50K | 76.61 | 9.75 | 16.45 | 0.3797 | 0.4569 |
| | | 100K | 50.83 | 8.24 | 26.90 | 0.4854 | 0.5543 |
| | | 150K | 40.60 | 7.53 | 35.35 | 0.5268 | 0.5894 |
| | | 200K | 35.00 | 7.22 | 42.42 | 0.5525 | 0.5981 |
| | | 250K | 31.77 | 7.11 | 47.22 | 0.5665 | 0.6117 |
| | | 300K | 29.26 | 7.03 | 51.62 | 0.5705 | 0.6141 |
| | | 350K | 27.34 | 6.94 | 55.10 | 0.5855 | 0.6127 |
| | | 400K | **26.21** | **6.86** | **58.04** | **0.5908** | **0.6112** |
| FCDM-L | 48.30 | 50K | 55.15 | 8.33 | 22.74 | 0.4655 | 0.5403 |
| | | 100K | 32.03 | 7.10 | 43.20 | 0.5791 | 0.5857 |
| | | 150K | 23.88 | 6.47 | 58.51 | 0.6106 | 0.6052 |
| | | 200K | 19.97 | 6.17 | 69.19 | 0.6312 | 0.6128 |
| | | 250K | 17.33 | 5.99 | 77.79 | 0.6427 | 0.6233 |
| | | 300K | 15.98 | 5.82 | 84.28 | 0.6477 | 0.6245 |
| | | 350K | 14.95 | 5.82 | 87.55 | 0.6527 | 0.6257 |
| | | 400K | **13.83** | **5.65** | **93.31** | **0.6612** | **0.6218** |
| FCDM-XL | 64.60 | 50K | 51.00 | 8.31 | 24.37 | 0.4940 | 0.5475 |
| | | 100K | 27.23 | 6.86 | 49.52 | 0.6108 | 0.5824 |
| | | 150K | 19.25 | 6.15 | 68.62 | 0.6492 | 0.5995 |
| | | 200K | 15.54 | 5.95 | 81.40 | 0.6690 | 0.6051 |
| | | 250K | 13.50 | 5.74 | 91.74 | 0.6785 | 0.6117 |
| | | 300K | 12.31 | 5.64 | 98.58 | 0.6829 | 0.6192 |
| | | 350K | 11.19 | 5.54 | 104.86 | 0.6914 | 0.6227 |
| | | 400K | **10.72** | **5.47** | **108.04** | **0.6864** | **0.6273** |

Table 7: **Performance of FCDMs across scales and training steps on ImageNet 256×256 (Diffusion).** Scaling FCDMs consistently leads to improved generative performance when trained with the diffusion objective.

| Model | FLOPs (G) | Training Steps | FID ↓ | sFID ↓ | IS ↑ | Precision ↑ | Recall ↑ |
|---|---|---|---|---|---|---|---|
| FCDM-S | 3.10 | 50K | 103.10 | 13.10 | 12.23 | 0.2863 | 0.3111 |
| | | 100K | 76.95 | 11.12 | 16.96 | 0.3826 | 0.4547 |
| | | 150K | 66.97 | 10.18 | 20.56 | 0.4228 | 0.4953 |
| | | 200K | 60.53 | 9.62 | 23.90 | 0.4428 | 0.5372 |
| | | 250K | 55.94 | 9.53 | 26.16 | 0.4670 | 0.5515 |
| | | 300K | 52.43 | 9.22 | 28.78 | 0.4787 | 0.5632 |
| | | 350K | 49.76 | 9.05 | 31.06 | 0.4866 | 0.5730 |
| | | 400K | **47.84** | **8.91** | **32.89** | **0.4944** | **0.5749** |
| FCDM-B | 12.20 | 50K | 80.10 | 18.48 | 15.46 | 0.3286 | 0.4072 |
| | | 100K | 52.23 | 8.34 | 25.95 | 0.4891 | 0.5450 |
| | | 150K | 42.58 | 7.85 | 33.83 | 0.5346 | 0.5780 |
| | | 200K | 37.01 | 7.51 | 40.72 | 0.5554 | 0.5819 |
| | | 250K | 33.14 | 7.24 | 46.39 | 0.5728 | 0.5855 |
| | | 300K | 30.16 | 7.07 | 51.60 | 0.5883 | 0.5953 |
| | | 350K | 28.40 | 6.99 | 55.07 | 0.5941 | 0.6066 |
| | | 400K | **26.61** | **6.85** | **58.51** | **0.6050** | **0.6017** |
| FCDM-L | 48.30 | 50K | 57.32 | 15.82 | 21.61 | 0.4467 | 0.4872 |
| | | 100K | 33.71 | 7.74 | 40.77 | 0.5852 | 0.5615 |
| | | 150K | 26.10 | 6.94 | 54.79 | 0.6232 | 0.5865 |
| | | 200K | 21.91 | 6.63 | 65.12 | 0.6429 | 0.5909 |
| | | 250K | 19.39 | 6.47 | 73.55 | 0.6557 | 0.5940 |
| | | 300K | 17.59 | 6.31 | 80.47 | 0.6650 | 0.6075 |
| | | 350K | 16.27 | 6.18 | 85.62 | 0.6681 | 0.6064 |
| | | 400K | **15.30** | **6.17** | **90.09** | **0.6751** | **0.6126** |
| FCDM-XL | 64.60 | 50K | 51.21 | 11.89 | 24.21 | 0.5010 | 0.5006 |
| | | 100K | 29.37 | 7.34 | 46.27 | 0.6172 | 0.5680 |
| | | 150K | 22.07 | 6.77 | 63.02 | 0.6572 | 0.5870 |
| | | 200K | 17.98 | 6.34 | 75.71 | 0.6747 | 0.5909 |
| | | 250K | 15.72 | 6.24 | 85.30 | 0.6853 | 0.5991 |
| | | 300K | 14.16 | 6.07 | 92.79 | 0.6952 | 0.6040 |
| | | 350K | 13.06 | 5.97 | 98.22 | 0.6976 | 0.6072 |
| | | 400K | **12.11** | **5.96** | **103.07** | **0.7030** | **0.6070** |

Table 8: **Performance of FCDMs across scales and training steps on ImageNet 256×256 (Flow-Matching).** Scaling FCDMs also demonstrates consistent improvements in generative performance when trained with the flow-matching objective.

## F  FREQUENCY-BASED ANALYSIS

To better highlight the differences between our fully convolutional diffusion model (FCDM) and the transformer-based DiT, we examine the evolution of the spectral energy—defined as the sum of the log-magnitude spectrum of the predicted noise—over the course of the diffusion process (using models trained for 400k iterations at 512x512 resolution). For each predicted noise sample, we compute the 2D Fourier transform, take the magnitude spectrum, and apply a logarithmic scaling $\log(1 + F)$ to compress the dynamic range. We then define the total spectral energy as the sum of all values in this log-magnitude spectrum, which reflects the overall distribution of frequency components. Figure 7 presents the total spectral energy, averaged over 128 validation samples, calculated at each of the 1,000 diffusion timesteps.

Across all diffusion steps, FCDM consistently exhibits higher spectral energy than DiT. This difference is most pronounced in the early-to-middle stages of the diffusion trajectory, where the model must simultaneously capture global structure and fine-grained detail. The elevated energy of FCDM indicates that its predicted noise retains stronger high-frequency components, which can be associated with sharper textures, edges, and local structures. By contrast, DiT produces lower spectral energy, suggesting smoother predictions with fewer high-frequency details. While this observation may provide a partial explanation for the performance gap between FCDM and DiT, further theoretical analysis is required.

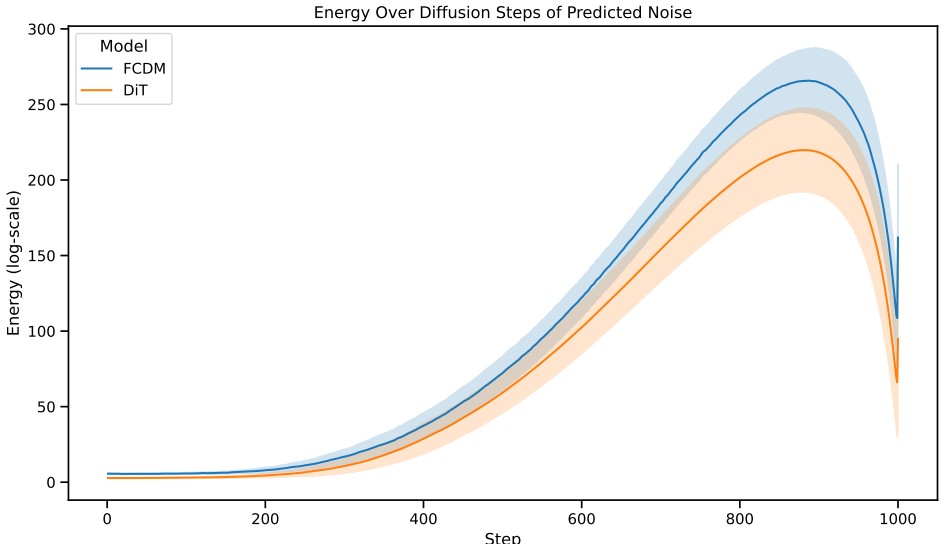

Figure 7: **Spectral energy of predicted noise across diffusion steps.** FCDM consistently exhibits higher spectral energy than DiT across the entire diffusion process, suggesting potential for better preservation of high-frequency components.

# G ANALYSIS OF ADDITIONAL ARCHITECTURAL VARIANTS

This section provides further details on the ablations in the main manuscript and introduces additional architectural ablation results.

Table 5 presents architectural ablations using the Large (L) model on ImageNet at $256\times256$. We analyze the effect of the contributions of specific architectural elements, including kernel size, FCDM blocks, Global Response Normalization (GRN), and U-shaped structure. Interestingly, although ConvNeXt (Liu et al., 2022; Woo et al., 2023) was developed for a different task, we observe similar ablation trends in our experiments.

**Large Kernel Sizes.** Vision Transformers employ non-local self-attention, enabling each layer to access a global receptive field. In contrast, ConvNets traditionally relied on stacking small $3\times3$ convolutions (popularized by VGGNet (Simonyan & Zisserman, 2015)), which are efficient on modern GPUs (Lavin & Gray, 2016). Our experiments show that reducing kernel sizes consistently degrades performance, with FID increasing from 19.97 to 20.48 and 21.28, indicating that larger kernels strengthen local operations and better approximate a global receptive field.

**Effect of FCDM Blocks.** We replaced FCDM blocks with ResNet blocks (He et al., 2016) using standard $3\times3$ convolutions. To match FLOPs, the hidden channels were reduced from 512 to 336, given the higher computational cost of standard convolutions compared to separable convolutions. This substitution results in a substantial degradation, with FID increasing from 19.97 to 31.14, indicating that the FCDM block is better suited for this task than the ResNet block.

**Effect of Architectural Design.** Most convolutional networks adopt a U-shaped design with skip connections, which facilitate integration of global and local features. This design helps preserve high-resolution detail while capturing overall context, and our model benefits similarly. Reshaping FCDM into an isotropic architecture, without downsampling and with constant resolution across depths, severely degrades performance, with FID increasing from 19.97 to 41.15. This indicates that the U-shaped architecture contributes to the performance gains of FCDM.

**Effect of GRN.** Subsequent work on ConvNeXt (Woo et al., 2023) reported that the original ConvNeXt (Han et al., 2022) suffered from feature collapse due to redundant channel activations. This issue was addressed by introducing Global Response Normalization (GRN), which normalizes channel activations. Since convolutional architectures can exhibit similar channel redundancy when applied to image generation (Ai et al., 2025), we likewise incorporate GRN into FCDM to mitigate redundant activations. Removing GRN from our model leads to a clear performance drop, with FID increasing from 19.97 to 21.24, highlighting its importance. This observation can also be clearly seen in the feature activation visualization of Figure 8. By reducing channel redundancy, GRN facilitates more balanced and diverse utilization of feature representations.

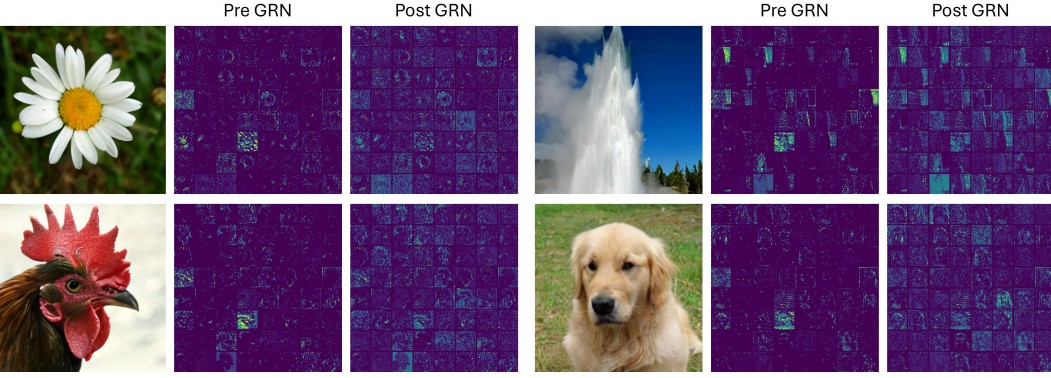

Figure 8: **Feature activation visualization.** We visualize the features before and after the GRN layer during the sampling of each image shown on the left. The first 64 channels of the last block in the first stage are displayed as $8\times8$ grids. GRN clearly reduces channel redundancy.

| Model | Training iterations | FLOPs (G) ↓ | FID ↓ |
|---|---|---|---|
| *SD-VAE* | | | |
| DiT-XL/2 | 400K | 118.6 | 19.47 |
| FCDM-XL | 400K | **64.6** | **11.57** |
| *EQ-VAE* | | | |
| DiT-XL/2 | 400K | 118.6 | 14.50 |
| FCDM-XL | 400K | **64.6** | **10.72** |

Table 9: **Ablation study on autoencoders.** Across different latent spaces (SD-VAE and EQ-VAE), FCDM consistently outperforms DiT.

| Model Configuration | Hidden channel $C$ | Depths | Params (M) | FLOPs (G) ↓ | TP (it/s) ↑ | FID ↓ |
|---|---|---|---|---|---|---|
| *Asymmetric U-Net Ablations* | | | | | | |
| **FCDM-L** (Default: Sym. U-Net) | 512 | [2, 4, 8, 4, 2] | 504.5 | 48.3 | 381.3 | 19.97 |
| Asym. U-Net | 512 | [2, 3, 8, 5, 2] | 504.5 | **48.3** | **381.3** | **19.55** |
| **FCDM-XL** (Default: Sym. U-Net) | 512 | [3, 6, 12, 6, 3] | 698.8 | **64.6** | **272.7** | **15.54** |
| Asym. U-Net | 512 | [3, 3, 12, 9, 3] | 698.8 | 64.6 | 272.7 | 15.55 |
| Asym. U-Net | 512 | [3, 5, 12, 7, 3] | 698.8 | **64.6** | **272.7** | **15.54** |
| *Downsample Levels Ablations* | | | | | | |
| **FCDM-L** (Default: 3-lvl U-Net) | 512 | [2, 4, 8, 4, 2] | 504.5 | 48.3 | 381.3 | 19.97 |
| 2-lvl U-Net* | 1024 | [4, 8, 4] | 482.1 | 140.0 | 153.5 | **17.74** |
| 4-lvl U-Net* | 256 | [3, 3, 3, 8, 3, 3, 3] | 497.5 | **16.0** | **873.7** | 28.67 |

Table 10: **Model architecture ablation studies on ImageNet 256×256.** All models trained for 200k iterations under identical training settings. * indicates that the hidden channel is adjusted to match number of parameters.

**Effect of Autoencoders.** Since FCDM operates in latent space, we tested whether performance persists under different VAEs. As shown in Table 9, FCDM consistently outperforms DiT under both SD-VAE (Rombach et al., 2022) and EQ-VAE (Kouzelis et al., 2025). Similar to DiT, our model performs best with EQ-VAE, improving further over SD-VAE. These findings suggest that techniques originally proposed to enhance DiT (e.g., stronger VAEs) transfer equally well to FCDM, indicating the potential for further performance improvements.

**Asymmetric Encoder–Decoder Allocation.** Following (Hoogeboom et al., 2025), we investigated whether an asymmetric allocation of compute between the encoder and decoder could outperform the symmetric setup. Intuitively, assigning more compute to the decoder appears advantageous, since upsampling from low to high resolution is more demanding and additionally requires processing skip connections. As shown in Table 10, an asymmetric design slightly improves performance for the L-model (FID decreases from 19.97 to 19.55). However, for the XL-model, the asymmetric setup performs on par with the symmetric variant. While asymmetric encoder–decoder architectures remain an interesting direction, we adopt the symmetric setup for its simplicity and more straightforward scalability.

**Effect of U-Net Depth.** While our base model employs a 3-level U-Net, we examined the effect of varying U-Net depth by changing the number of downsampling levels. A 2-level U-Net reduces the FID of the Large (L) model from 19.97 to 17.74, but its FLOPs increase sharply to 140G, more than double those of the XL model, indicating that the modest FID gain does not justify the additional cost. In contrast, a 4-level U-Net significantly lowers FLOPs, since most computation is performed at the lowest resolution, but this comes at the expense of a higher FID. Although further exploration of deeper U-Nets may be worthwhile, the 3-level U-Net already provides a strong trade-off: it ensures a large global receptive field, especially when paired with the large kernels of the FCDM block, while maintaining a balanced parameter-to-FLOPs ratio and a scalable design.

**Block Scaling across U-Net Levels.** We conducted an additional ablation study on how to distribute the number of blocks across U-Net levels as resolution decreases, while keeping the total number of blocks fixed. Specifically, we compared three scaling strategies: quadratic scaling (default, where the number of blocks increases quadratically with depth), constant allocation (the same number of blocks at each level), and linear scaling (the number of blocks increases linearly with

| Model Configuration | Hidden channel $C$ | Depths | Params (M) | FLOPs (G) ↓ | TP (it/s) ↑ | FID ↓ |
|---|---|---|---|---|---|---|
| **FCDM-S** (Default: Quadratic Scaling) | 128 | [2, 4, 8, 4, 2] | 32.7 | **3.1** | **2687.2** | 60.33 |
| Constant | 144 | [4, 4, 4, 4, 4] | 29.9 | 3.7 | 1672.4 | **59.57** |
| Linear Scaling | 136 | [2, 4, 6, 4, 2] | 31.4 | 3.1 | 2418.7 | 61.16 |
| **FCDM-B** (Default: Quadratic Scaling) | 256 | [2, 4, 8, 4, 2] | 127.7 | **12.2** | **1001.6** | 35.00 |
| Constant | 296 | [4, 4, 4, 4, 4] | 122.6 | 15.4 | 647.2 | **32.75** |
| Linear Scaling | 272 | [2, 4, 6, 4, 2] | 122.5 | 12.3 | 922.3 | 36.27 |
| **FCDM-L** (Default: Quadratic Scaling) | 512 | [2, 4, 8, 4, 2] | 504.5 | **48.3** | **381.3** | 19.97 |
| Constant | 600 | [4, 4, 4, 4, 4] | 496.6 | 62.8 | 261.6 | **17.63** |
| Linear Scaling | 552 | [2, 4, 6, 4, 2] | 497.8 | 49.4 | 358.7 | 19.87 |

Table 11: **Ablation study on block scaling strategies.** Comparison of block scaling strategies across the S, B, and L models. The default quadratic scaling (number of blocks increasing quadratically) is compared with constant and linear variants, where the total number of blocks is fixed but redistributed across U-Net levels. All models are trained on ImageNet $256 \times 256$ for 200k iterations under identical settings. The hidden channel is adjusted to match the total number of parameters.

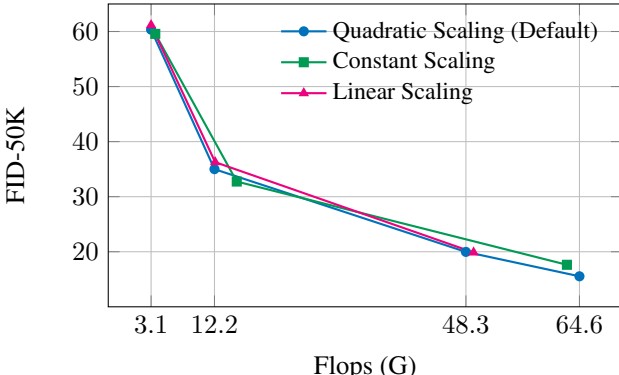

Figure 9: **Comparison of scaling strategies.** The plot compares three ways of distributing blocks across U-Net resolution levels. Allocating more blocks at lower resolutions can improve efficiency by concentrating computation where it is cheaper, although the gains remain modest.

depth). Comprehensive results are reported in Table 11, which summarizes FID, parameter counts, FLOPs, and throughput across the S, B, and L models under the three strategies. Overall, as shown in Figure 9, quadratic scaling provides a slightly better trade-off, yielding lower FID for a given amount of computation.

# H    MORE QUALITATIVE RESULTS

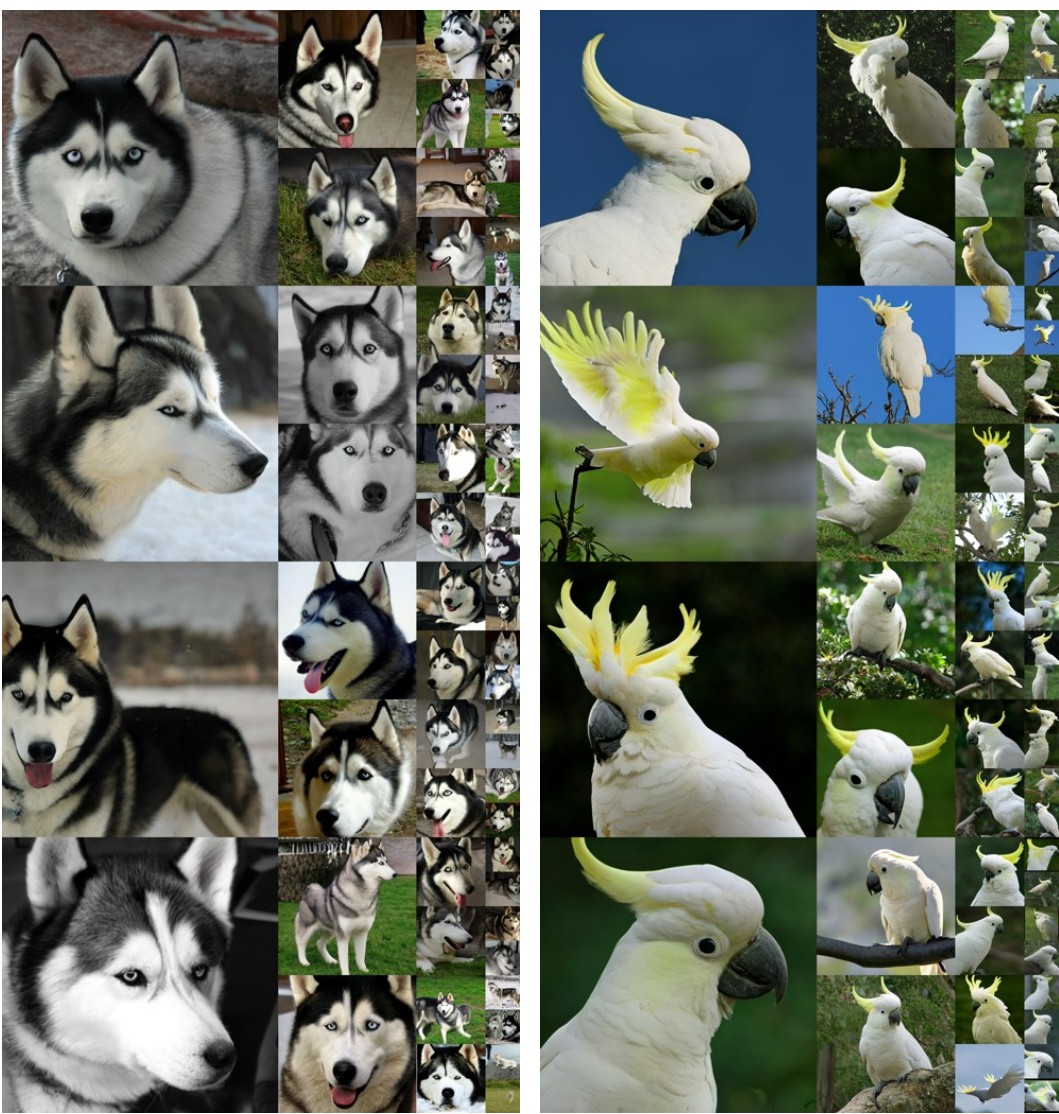

Class label = "husky" (250)          Class label = "sulphur-crested cockatoo" (89)

Figure 10: **Uncurated 512×512 FCDM-XL samples.** Classifier-free guidance scale = 4.0

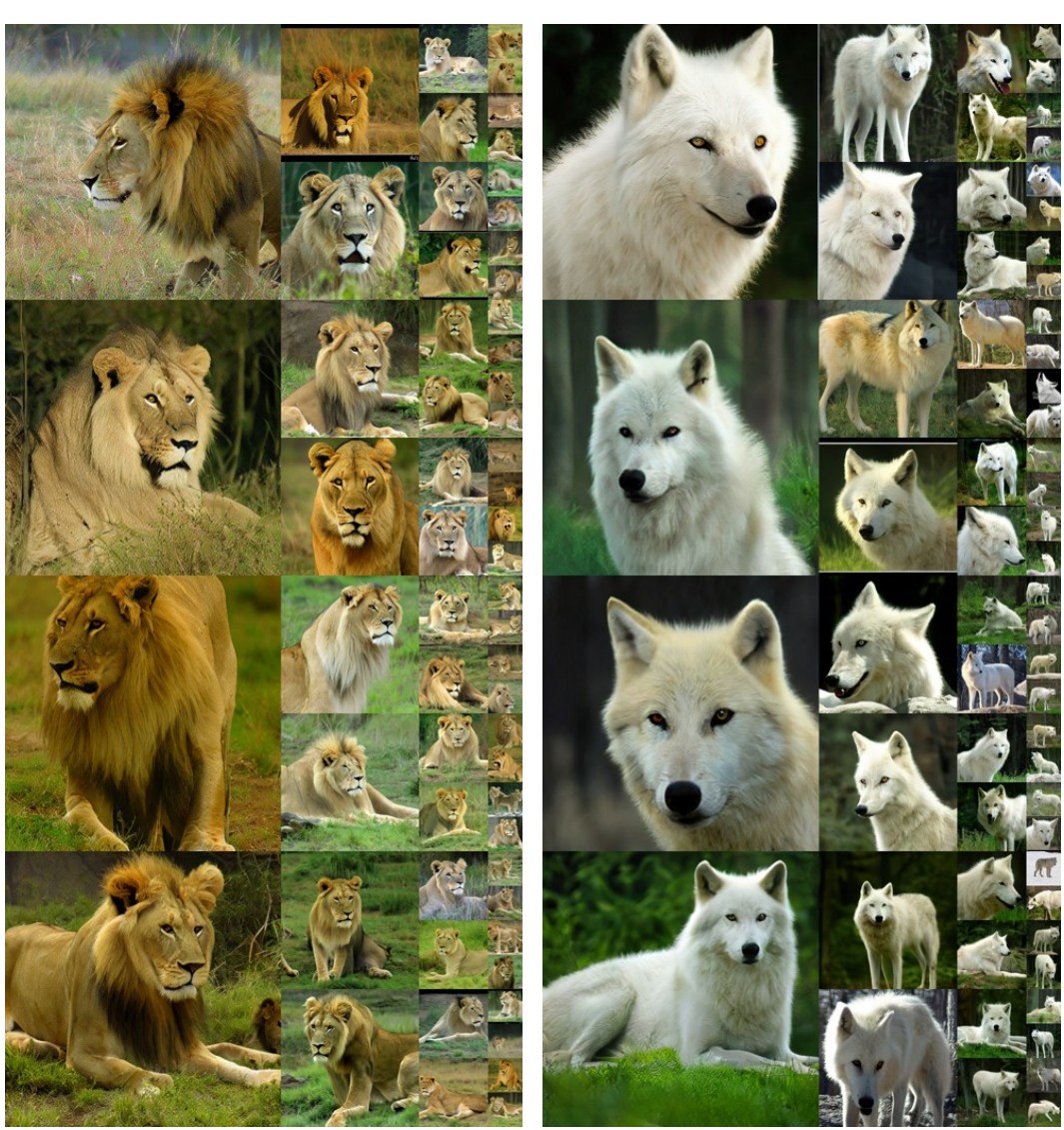

Class label = "lion" (291)                    Class label = "arctic wolf" (270)

Figure 11: **Uncurated 512×512 FCDM-XL samples.** Classifier-free guidance scale = 4.0

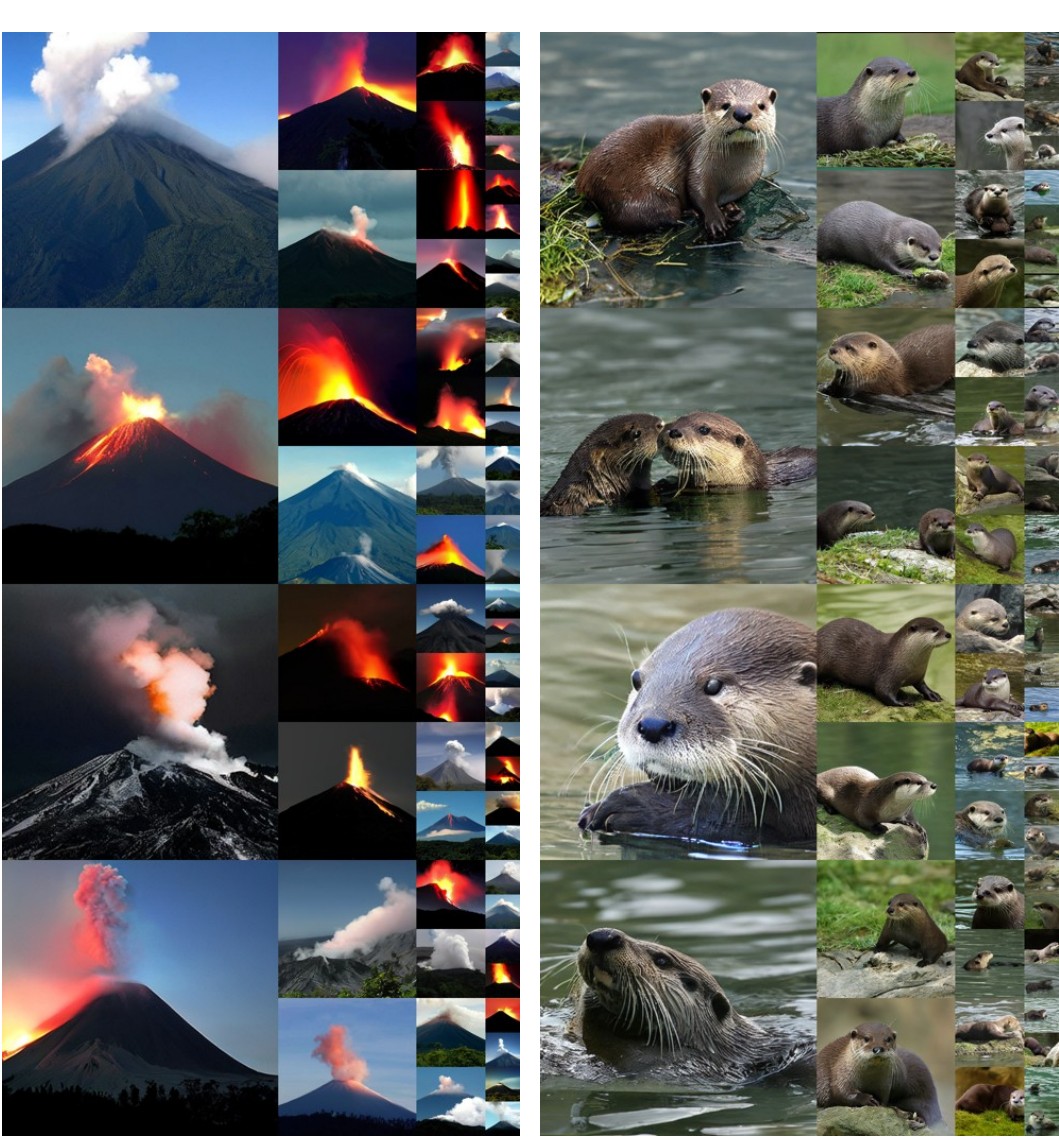

Class label = "volcano" (980)          Class label = "otter" (360)

Figure 12: **Uncurated 512×512 FCDM-XL samples.** Classifier-free guidance scale = 4.0

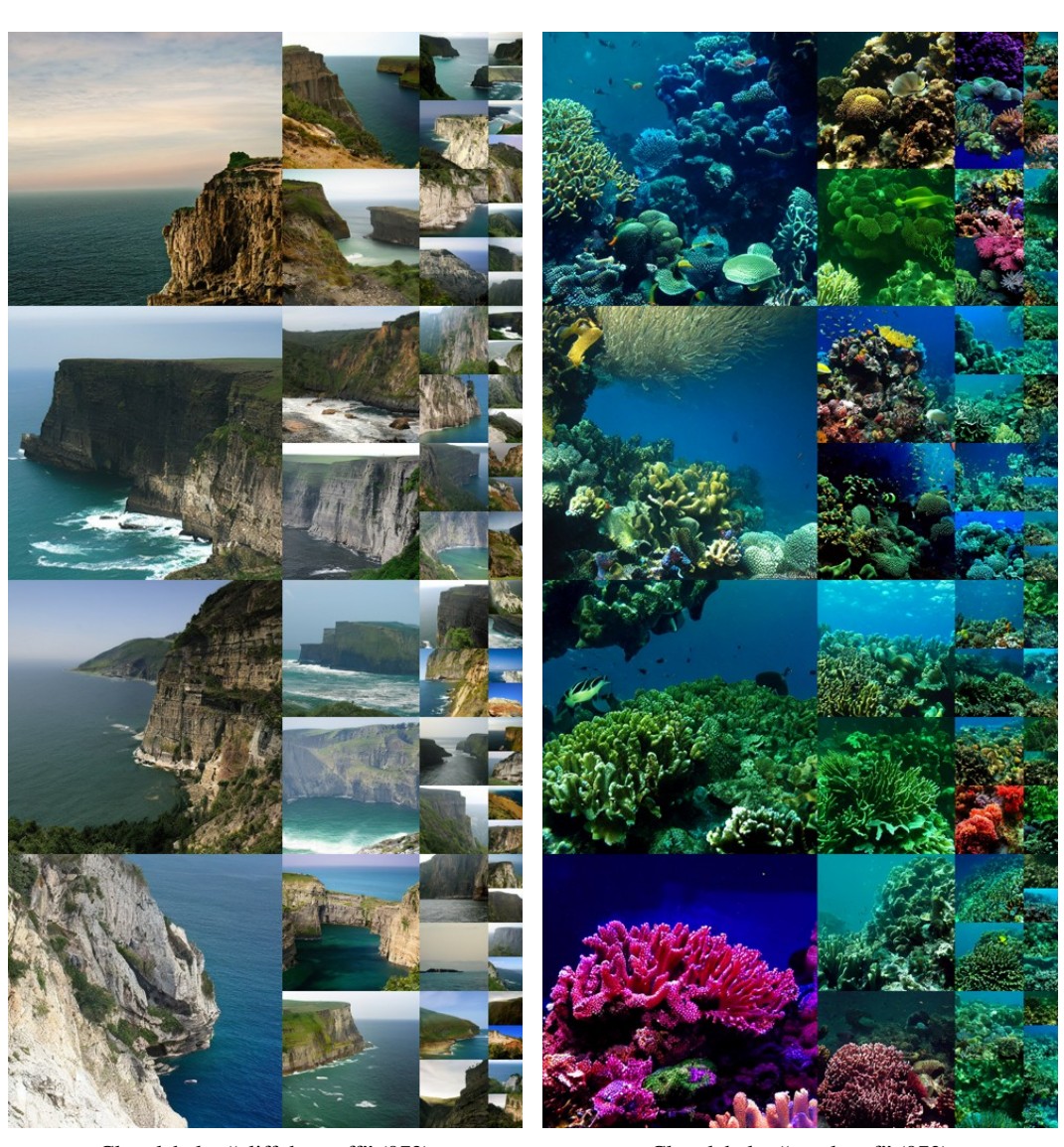

Class label = "cliff drop-off" (972)          Class label = "coral reef" (973)

Figure 13: **Uncurated 512×512 FCDM-XL samples.** Classifier-free guidance scale = 4.0

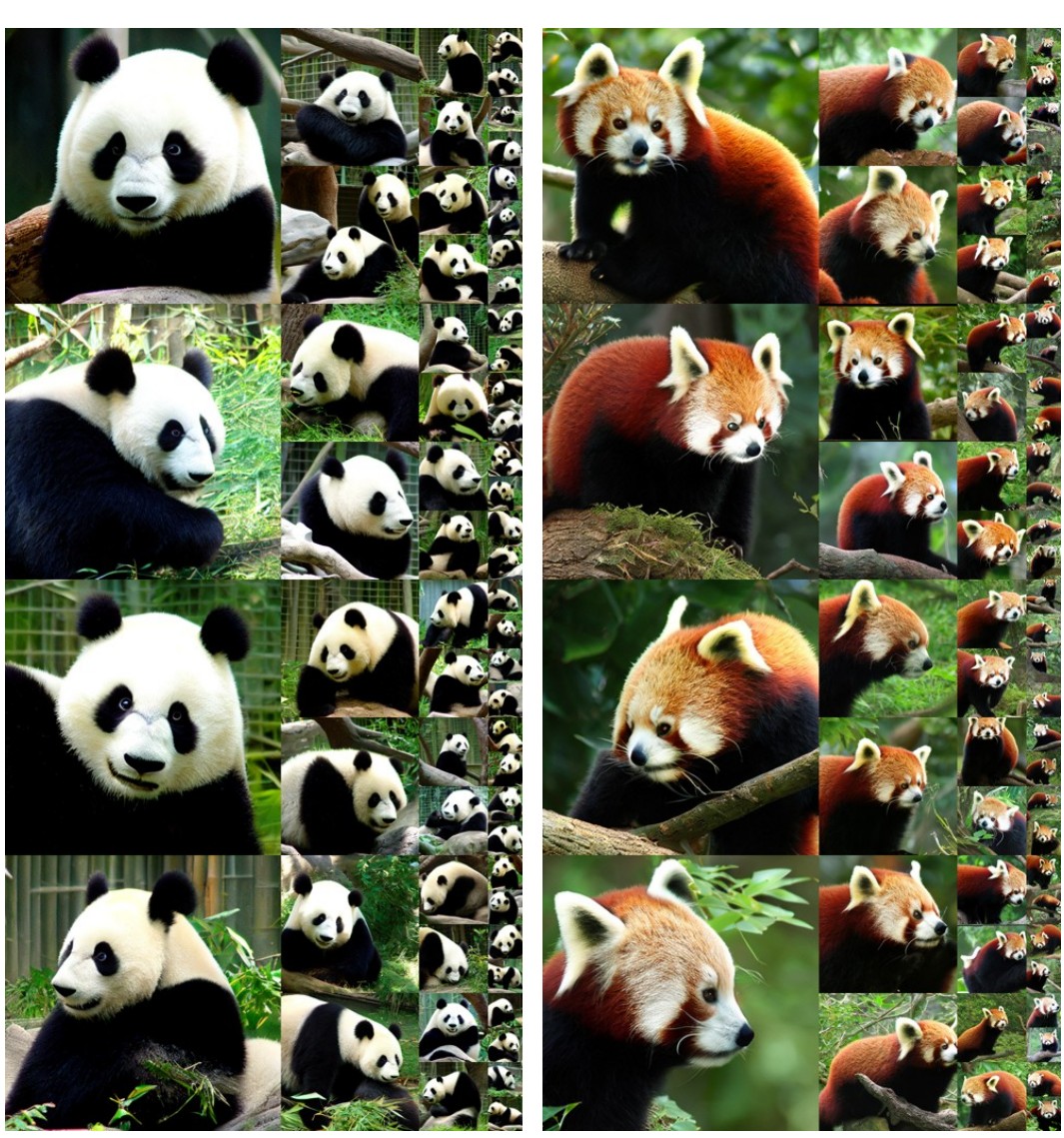

Class label = "panda" (388)          Class label = "red panda" (387)

Figure 14: **Uncurated 512×512 FCDM-XL samples.** Classifier-free guidance scale = 4.0

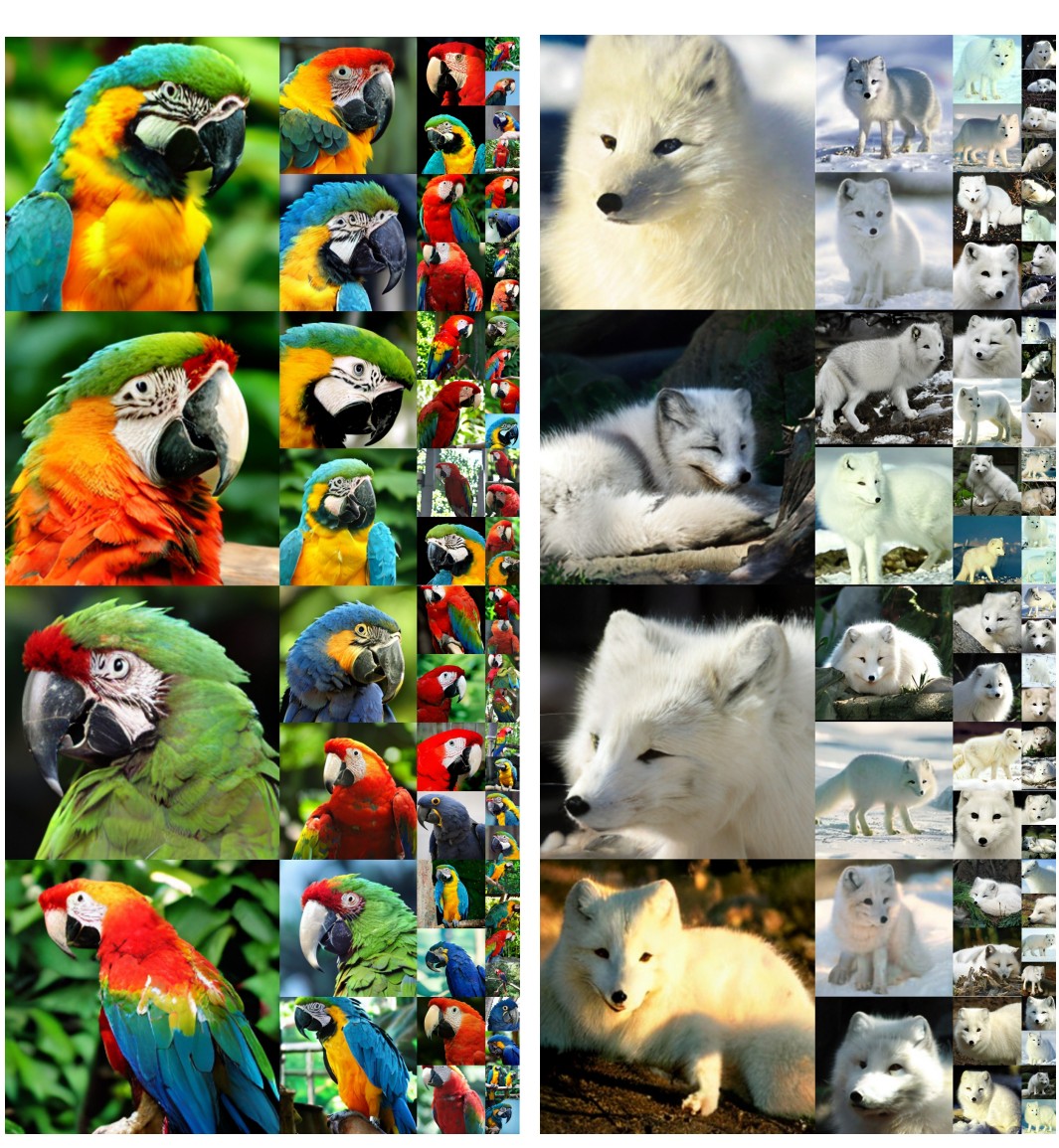

Class label = "macaw" (88)          Class label = "arctic fox" (279)

Figure 15: **Uncurated 256×256 FCDM-XL samples.** Classifier-free guidance scale = 4.0

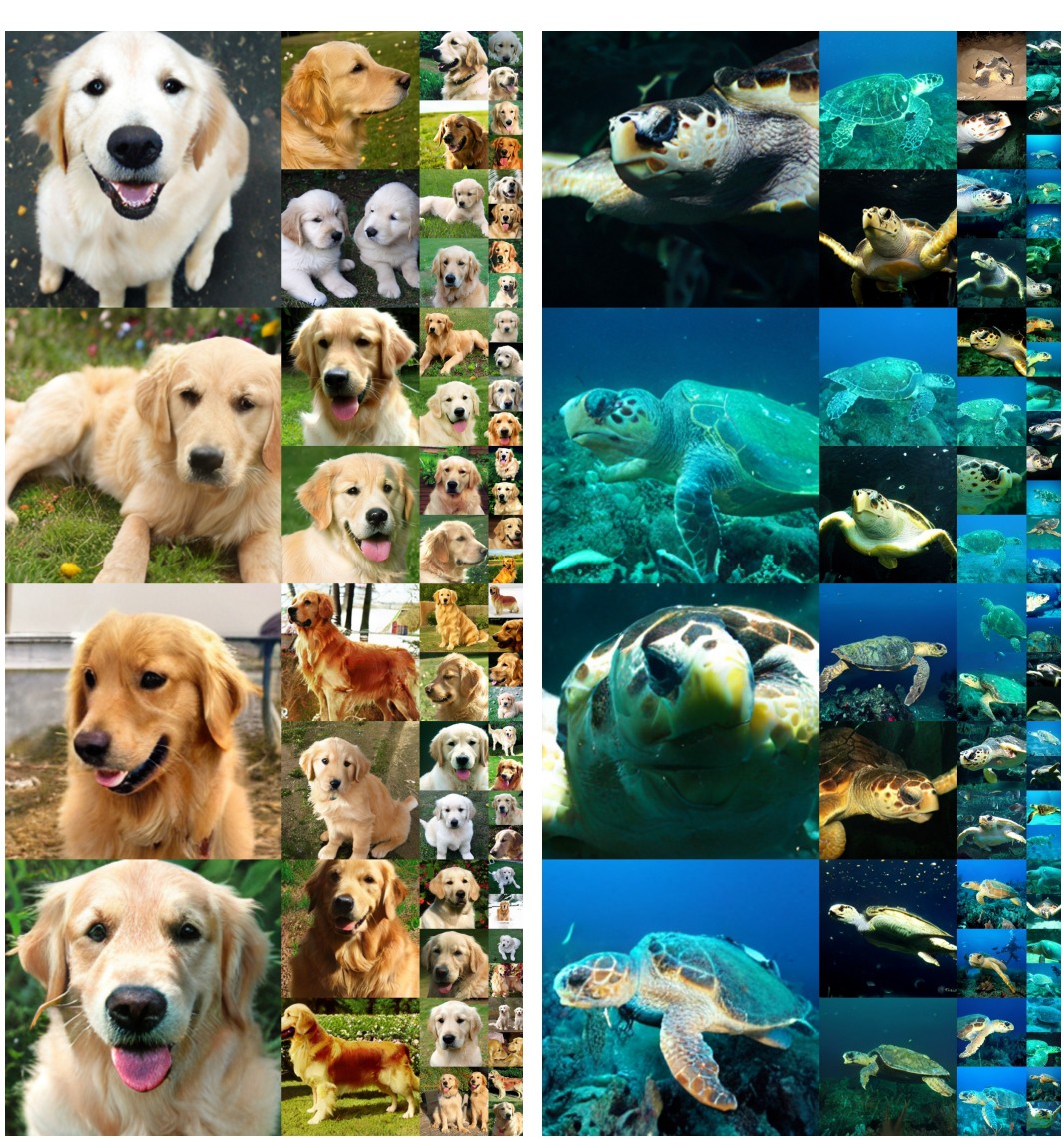

Class label = "golden retriever" (207)          Class label = "loggerhead sea turtle" (33)

Figure 16: **Uncurated 256×256 FCDM-XL samples.** Classifier-free guidance scale = 4.0

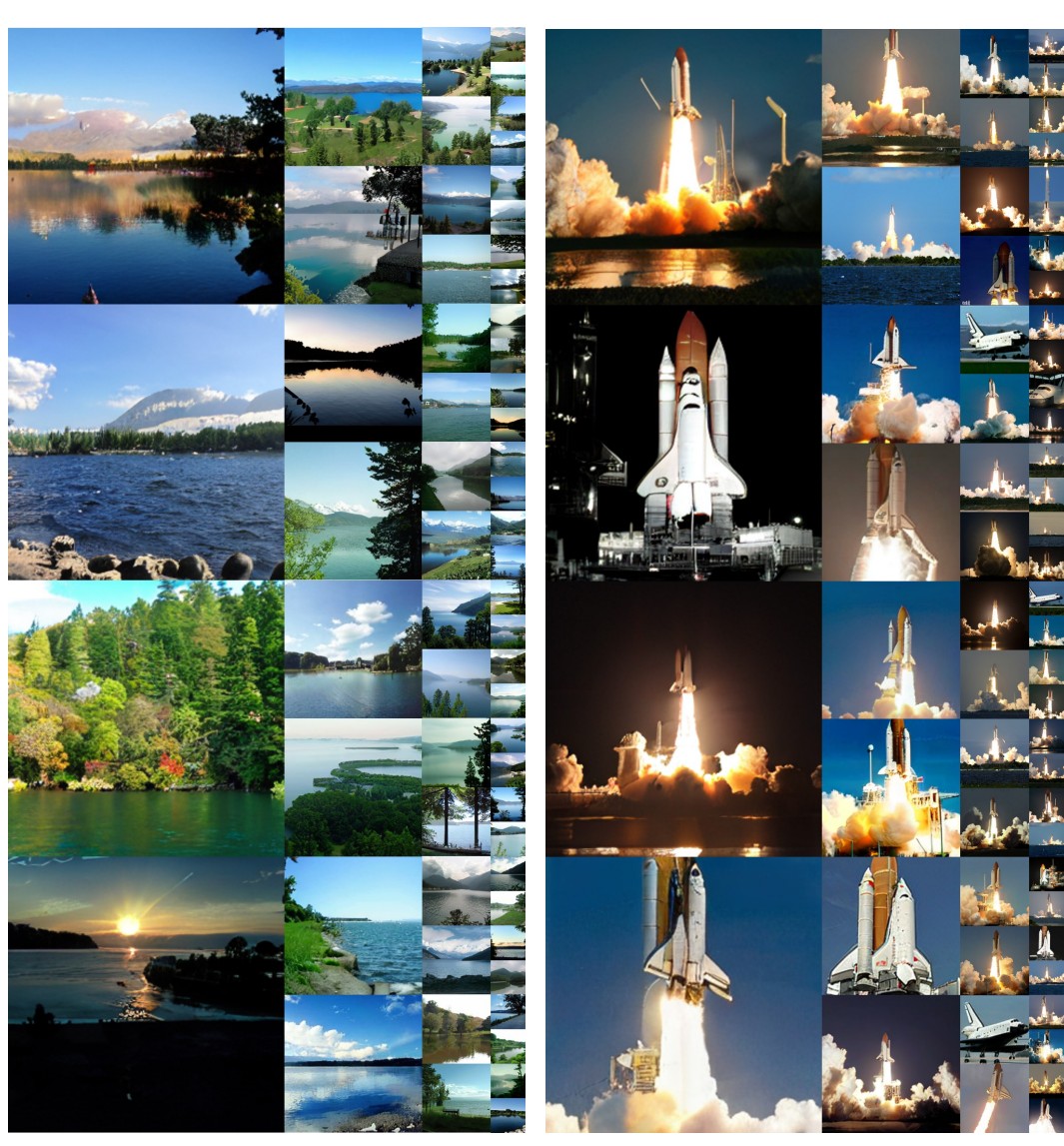

Class label = "lake shore" (975)          Class label = "space shuttle" (812)

Figure 17: **Uncurated 256×256 FCDM-XL samples.** Classifier-free guidance scale = 4.0

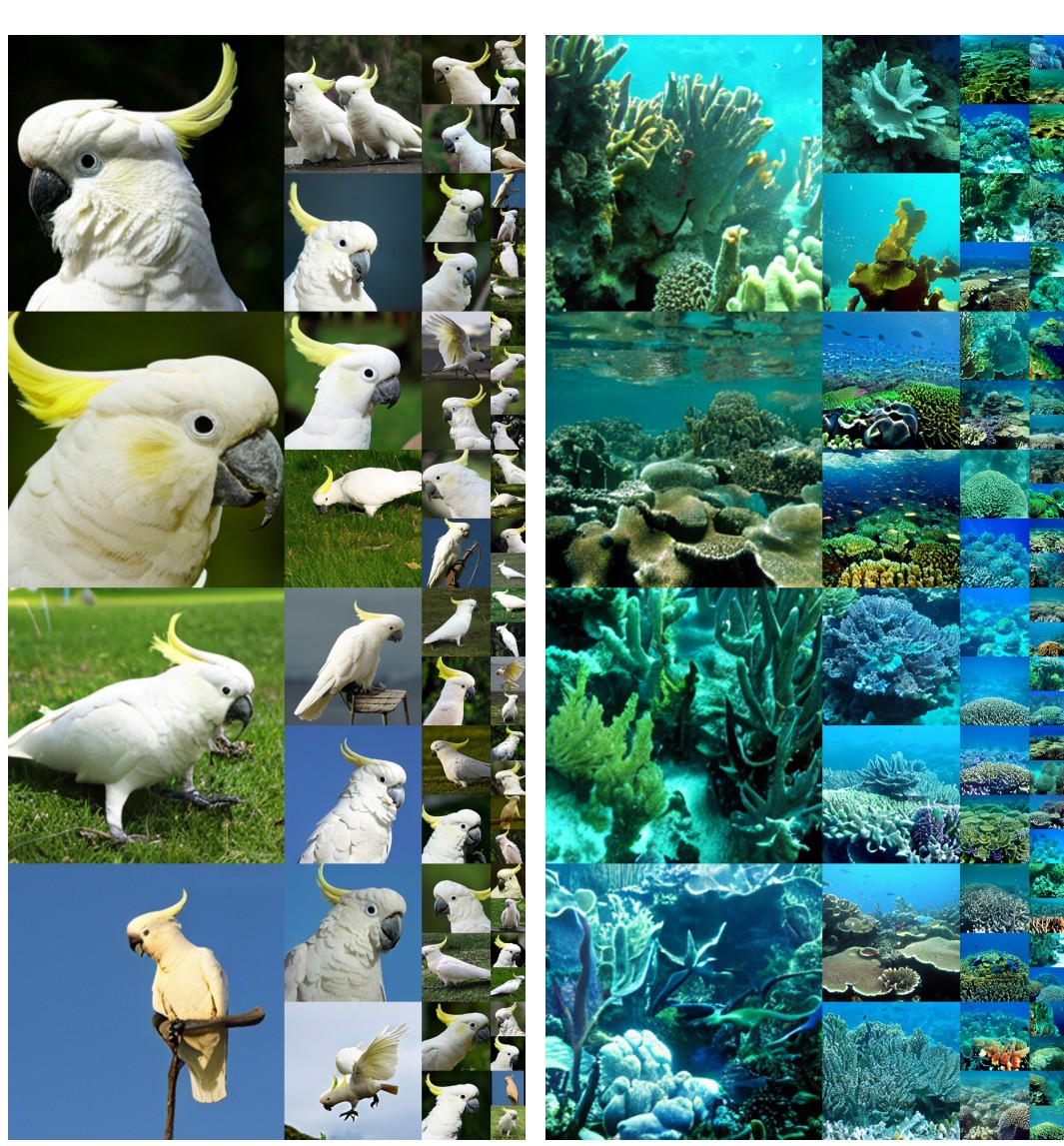

Class label = "sulphur-crested cockatoo" (89)        Class label = "coral reef" (973)

Figure 18: **Uncurated 256×256 FCDM-XL samples.** Classifier-free guidance scale = 4.0

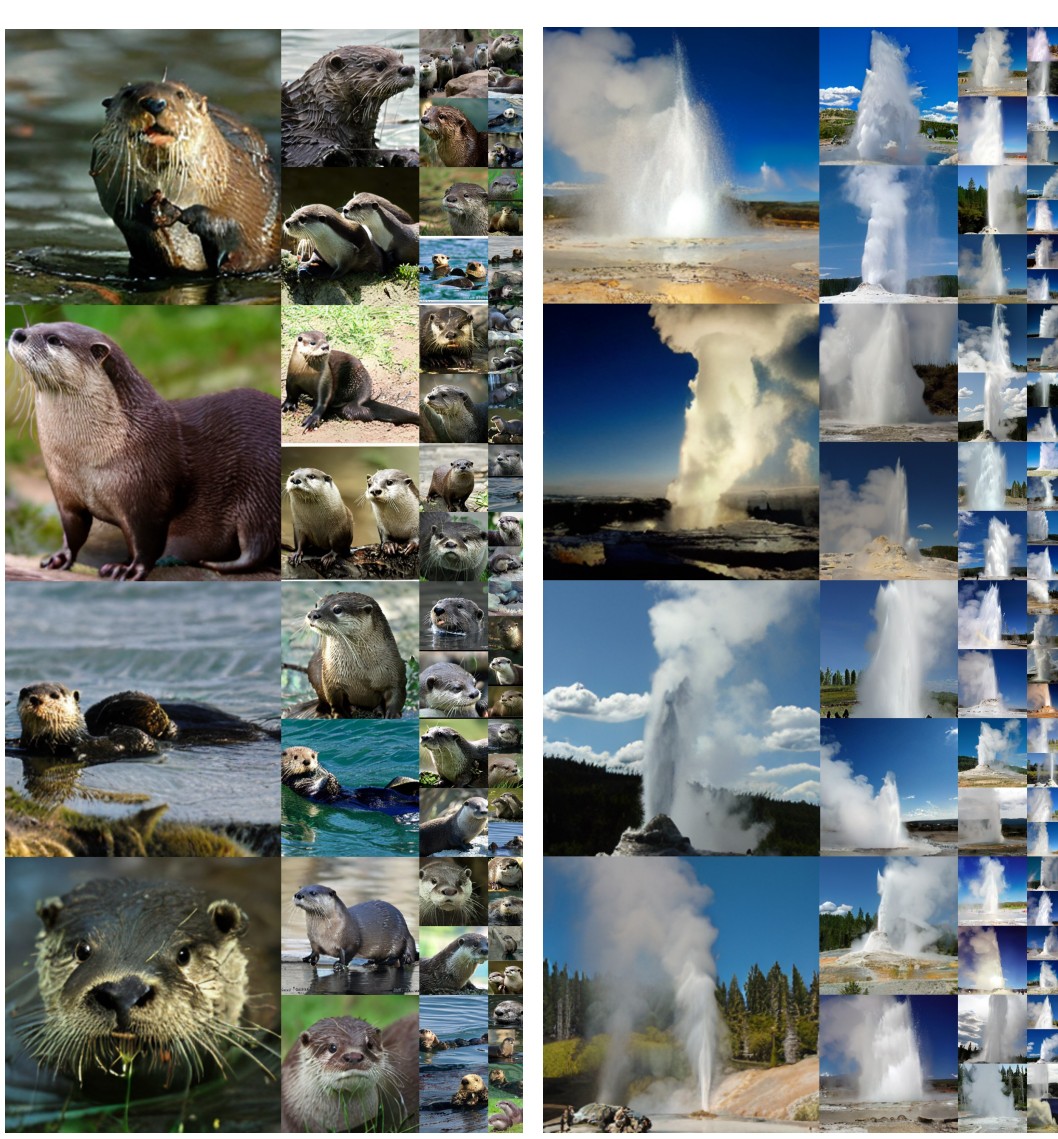

Class label = "otter" (360)          Class label = "geyser" (974)

Figure 19: **Uncurated 256×256 FCDM-XL samples.** Classifier-free guidance scale = 4.0

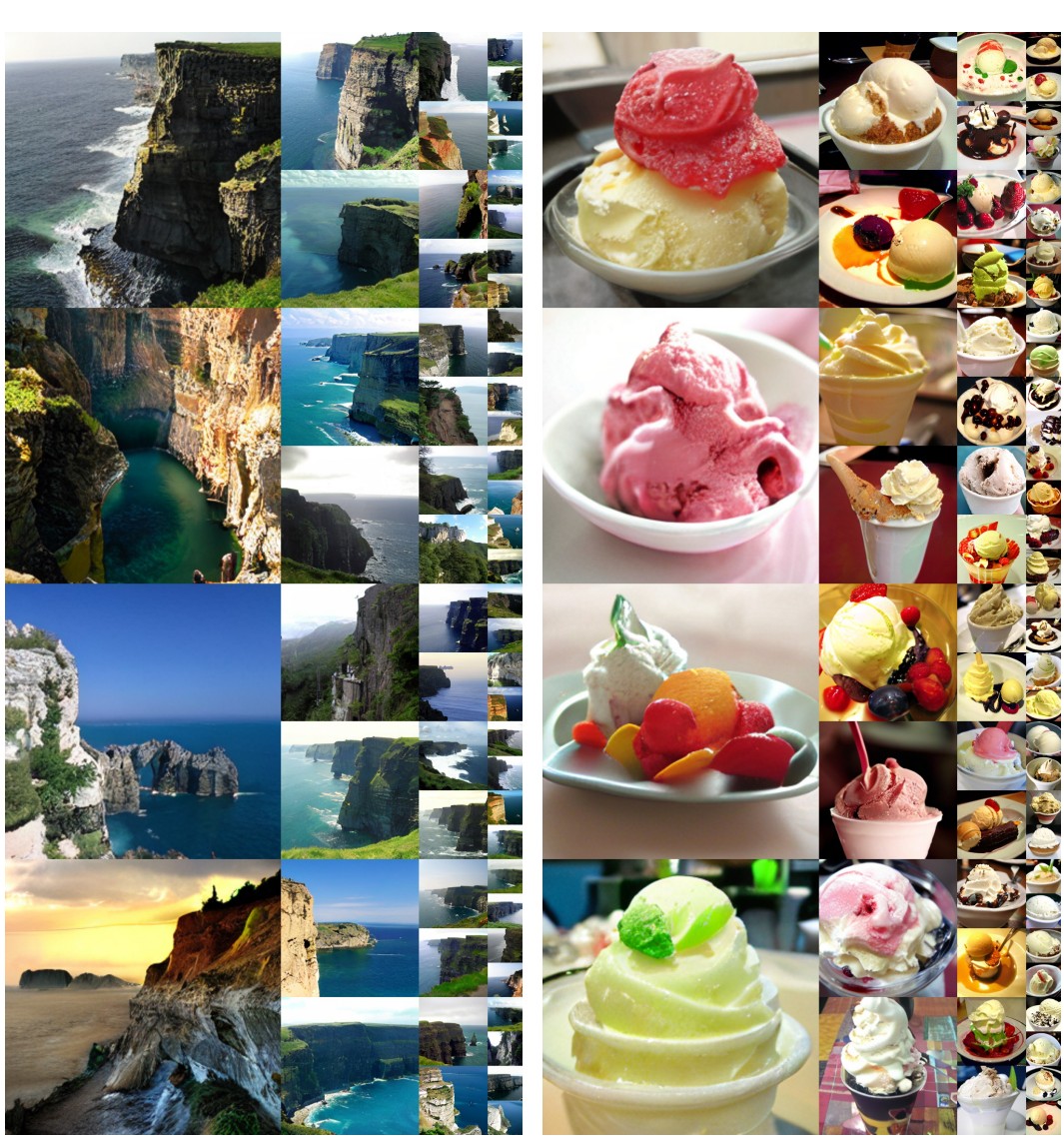

Class label = "cliff drop-off" (972)                    Class label = "ice cream" (928)

Figure 20: **Uncurated 256×256 FCDM-XL samples.** Classifier-free guidance scale = 4.0

## I  FUTURE RESEARCH DIRECTIONS

**Integration with DINOv3.**   Interestingly, DINOv3 (Siméoni et al., 2025) was recently released, along with ConvNeXt-based distilled weights. Given the architectural similarity, integrating such powerful self-supervised visual representations (e.g., REPA (Yu et al., 2025)) into our framework would be an important direction for future work.

**Different input data types.**   While this work focuses exclusively on class-conditional image generation on ImageNet, we do not investigate its applicability to text-to-image generation tasks. Transformers are naturally suited for multimodality through mechanisms such as cross-attention, whereas achieving similar cross-modal integration using only convolutional operations remains challenging. Exploring alternative strategies for enabling multimodality within purely convolutional architectures would therefore be an interesting future direction. Moreover, leveraging the efficiency of convolutional networks to extend generative modeling toward high-dimensional data (e.g., video, 3D, or 4D) represents another promising avenue for future research.

**Scaling and training frameworks.**   An important future direction is to investigate how scaling up FCDM to larger model sizes (e.g., billions of parameters) influences its expressive power and sample quality, and how advanced training frameworks can be employed to improve model performance and accelerate training. Such efforts hold the potential to enable FCDM to achieve, or even surpass, state-of-the-art results while preserving its computational advantages.

**Theoretical analysis.**   Exploring in-depth theoretical insights into why ConvNeXt-based architectures work well will also be an exciting future direction. For example, it would be interesting to investigate whether the ConvNeXt, as a universal approximator, possesses expressive power equivalent or comparable to that of attention mechanisms.

