# OpenReview forum: "All Convolution, No Attention: Designing Diffusion with Convolutions"
_ICLR.cc/2026/Conference — ICLR 2026 Conference Withdrawn Submission_

### Official Review · Reviewer_yd2h · 2025-10-26

**Soundness:** 2
**Presentation:** 2
**Contribution:** 1
**Rating:** 2
**Confidence:** 4

**Summary:**

The paper presents FCDM, a fully convolutional backbone for diffusion models based on ConvNeXt and the AdaLN from DiT. A U-Net architecture is used. Experiments demonstrate that FCDM achieves competitive performance with substantially higher throughput compared to baselines.

**Strengths:**

-  FCDM achieves competitive FID with substantially higher throughput.
-  Ablation studies on model architecture details the effect of the architecture design

**Weaknesses:**

Limited novelty and contribution:
- This work is composition of existing methods and is highly similar to DiCo, which is also a fully convolutional diffusion model. The difference between this work and DiCo is the replacement of convolutions in DiCo with the operations in ConvNeXt. The core components of the model (ConvNeXt block, AdaLN, U-Net architecture) are all estabilished in previous works. I do not see efforts on exploring how convolutions can be adapted for generative models.
- FID Results are very close to DiCo (Table 3, Table 4)

Lack of experiments and baselines:
- Table 4 only presents ImageNet-1k 512x512 results without classifier-free guidance (CFG). Results with CFG should be reported.
- This paper should compare with other works exploring different model architectures for diffusion models (e.g. $D^2iT$, MDT). The performance is not on par with more advanced diffusion transformers.
- The paper claims in the introduction and Section 3 that depthwise convolutions are sufficient to outperform local attention. There should be experiments replacing depthwise convolution with local attention (window attention in Swin Transformer and neighborhood attention in NAT) to comfirm this.


References:

DiCo: Revitalizing ConvNets for Scalable and Efficient Diffusion Modeling. NeurIPS 2025

$D^2$iT: Dynamic Diffusion Transformer for Accurate Image Generation. CVPR 2025

MDTv2: Masked Diffusion Transformer is a Strong Image Synthesizer. https://arxiv.org/pdf/2303.14389

Swin Transformer: Hierarchical Vision Transformer using Shifted Windows. ICCV 2021

Neighborhood Attention Transformer. CVPR 2023

**Questions:**

- What are the advantages and disadvantages of convolution over attention in diffusion model architectures? Ablations in this paper demonstrate that larger kernel sizes lead to better performance, which is consistent with previous findings that global understanding is crucial for image generation. Using pure convolution without global receptive field likely leads to sub-optimal performance. I hope the authors can consider this question carefully and revise their work for publication.

---

### Official Review · Reviewer_en2K · 2025-10-29

**Soundness:** 3
**Presentation:** 3
**Contribution:** 3
**Rating:** 4
**Confidence:** 3

**Summary:**

The paper proposes a convolution-only architecture for image diffusion generative models, inspired by ConvNeXt design principles. It demonstrates competitive or superior performance to diffusion transformers on ImageNet 256x256 and 512x512, achieving higher image quality with lower FLOPs. The authors argue that depthwise convolutions provide local connectivity patterns similar to attention, making them an efficient alternative.

**Strengths:**

- Strong empirical results on large-scale benchmarks (ImageNet 256, 512) with clear computational efficiency gains.
- Simple and elegant design based on well-established convolutional backbones, improving interpretability and ease of implementation.
- Clear comparison to transformer-based diffusion models, showing that attention is not strictly necessary for high-quality generation.

**Weaknesses:**

- My main concern is the limited novelty justification. Especially, the conceptual difference from DiCo (NeurIPS 2025) is not clearly articulated (other than using ConvNeXt).
- The claim that depthwise convolution matches local attention lacks theoretical or empirical support beyond citation.
- Conditioning limitation: experiments focus only on class-conditional generation, leaving unclear whether the approach generalizes to text-to-image or multi-modal conditioning.

**Questions:**

see weaknesses

---

### Official Review · Reviewer_6KUW · 2025-10-31

**Soundness:** 4
**Presentation:** 4
**Contribution:** 2
**Rating:** 6
**Confidence:** 4

**Summary:**

This paper proposes a Fully Convolutional Diffusion Model (FCDM) that is free of the attention mechanism, which is one of the main component of recent diffusion model architecture. FCDM utilizes ConvNext blocks with LayerNorm replaced by AdaLN to incorporate time and class conditioning. FCDM shows superior generative quality on ImageNet 256x256 and 512x512 with fewer FLOPs and higher throughput.

**Strengths:**

- The paper is overall well-written and structurally sound.
- The quantitative results are impressive. FCDM shows very promising results on ImageNet generation tasks, considering the fact that attention is completely removed.
- FCDM does not require attention module, which has high overhead. This is surprising because the attention mechanism is the main success factor of the transformer-based model, and even the modern convolution-based model.
- The paper conducts various ablation studies to verify the necessity for each component in FCDM.

**Weaknesses:**

- FCDM can be viewed as combination DiCo with ConvNext, and the quantitative difference between DiCo and FCDM is also somewhat marginal.
- Current trend for generative models are multi-modality, where text are always involved. LMM must be able to process text and other modality in one sequence, yielding several advantages such as easier editing etc. Although effective in visual tasks, I am not convinced that FCDM can be extended to LMM, where it must process both text and other modalities.
- The performance is only tested on *class*-conditional task, which have only one semantic. I wonder if FCDM can handle text conditions where multiple semantics are presented.

**Questions:**

- In ablation study, only convolution window size smaller than 7x7. What if the window size is larger than 7?
- Attention allows fine-grain interaction between points and semantics in the conditioning signals. When only one semantic is presented, as in the class-conditional case, this fine-grain interaction might not be necessary. However, as mentioned in the Weakness, I wonder if this is the case when multiple semantics are presented in the condition, which is the usual case in text-to-image generation.
- To the best of my knowledge, attention or its variants was necessary component even for U-Net model with convolution blocks. What do you think is the main contribution of FCDM that allows the complete removal of the attention block?

---

### Official Review · Reviewer_URQh · 2025-11-01

**Soundness:** 3
**Presentation:** 3
**Contribution:** 2
**Rating:** 4
**Confidence:** 4

**Summary:**

The paper proposes a Fully Convolutional Diffusion Model (FCDM), which adapts the ConvNeXt architecture into a U-Net backbone for image generation. The authors argue that this fully convolutional approach challenges the current trend of using transformer-based backbones in diffusion models. They present experiments on ImageNet, showing that their FCDM models are more computationally efficient (fewer FLOPs, higher throughput) and achieve better FID scores compared to the Diffusion Transformer (DiT) across various model scales.

**Strengths:**

The paper presents a clear and valuable challenge to the prevailing view that Transformers are the only path forward for scaling diffusion models. The empirical results, within the scope of the comparison to DiT, are strong and demonstrate a compelling efficiency-performance trade-off. The proposed FCDM architecture is simple, scalable, and the experiments systematically ablate its core components, providing a solid foundation for the architectural choices made.

**Weaknesses:**

- The novelty is limited. The idea of "revisiting ConvNets for diffusion" is not new and draws upon previous works like DiCo and DiC exploring similar directions. While FCDM uses a ConvNeXt block, the overall contribution feels more incremental than foundational (X but with Y style paper). The paper compares against DiT but fails to compare against more principled U-Net-based architectures, such as the designs from the EDM/EDM-2 papers.

- The paper overstates the equivalence between its convolutional blocks and attention mechanisms. The authors draw an analogy between depthwise convolution and *local* (windowed) self-attention but conveniently ignore the role of *global* self-attention, which is widely considered the key component behind the powerful scaling and performance of transformers. CNNs are known to have limitations in modeling long-range spatial dependencies and capturing global context (eg. generating a globally coherent image of a wire mesh), a critical weakness that the paper does not address.

- The evaluation is weak and potentially misleading. The comparison is largely limited to DiT (which I would consider a very weak baseline in 2025), and the primary metric is FID. Recent studies have shown that FID has worse correlation with human perception compared to metrics based on more perceptually aligned encoders like DinoV2, which provide a richer and more reliable assessment of generative model performance. The absence of these more modern metrics calls the reported state-of-the-art claims into question, especially given the previous point about global structure.

- The proposed U-Net hyperparameter scaling rule (doubling channels and depth at each 2x downsampling stage) is presented as a simple and effective design choice, but it lacks a strong justification for its optimality. While the authors perform a basic ablation in Appendix G, the ablations suffer from poor FLOP matching and at the scales tested and metrics used I wouldn't say the result is anywhere near conclusive. DiT/SiT style architecture is relatively battle tested at this point and I would need to see better and larger scale ablations to be convinced this architecture is actually competitive.

- There are many papers released in the last couple of months that claim to accelerate diffusion transformer training by orders of magnitudes (REPA, TREAD, REDI[1], Contrastive flow matching...). To compellingly argue that a new architecture is superior to DiT, it is no longer sufficient to compare against a vanilla training recipe. It is crucial to demonstrate that the proposed architecture can also benefit from these SOTA training methods and maintains its advantage.


[1] https://arxiv.org/abs/2504.16064

**Questions:**

- Why are you not comparing against any "hybrid" unet+attention baselines?
- Why are you not using one of the more modern metrics? (eg. dinov2-fd or dino kernel distance etc)
- Can you conclusively show that having a fully convolutional model does not compromise long range relationships?
- Can you show experiments at larger scales (more steps and larger models)?
- A technique like ReDI should be directly and easily applicable to your approach as it doesn't require doing operations on the featuremaps of the models being trained. It should be also quick to test. Showing that the performance gap does not diminish/disappear vs. their baselines would make the paper stronger.
- Doing a simple sweep of basic hparams (eg. LR, WD, warm up, EMA..) for both the proposed model and baseline and presenting the optimal configuration of both would make the paper more convincing.

---

### Note · Authors · 2025-11-14

**Comment:**

We have decided to withdraw our submission, but we would like to sincerely thank all the reviewers for their constructive feedback, which has helped us improve our work.

**Withdrawal Confirmation:**

I have read and agree with the venue's withdrawal policy on behalf of myself and my co-authors.